# Position: Deciphering the Functions of DNAs, RNAs, and Proteins Should Consider Multi-Modal Large Language Models

**Pengtao Xie** [1]  **Victor Nizet** [1]  **Lei Wang** [2]  **Ahmed Alaa** [3]  **Daniel C. Zielinski** [1]  **Trey Ideker** [1]  **Bernhard Palsson** [1]

## Abstract

Understanding the functions of DNAs, RNAs, and proteins is fundamental to advancing life science research and enabling translational applications such as drug discovery and precision medicine. While deep learning methods have shown promise in biomolecular function prediction, they typically constrain outputs to predefined categories and require training separate models for each task. Existing multi-task learning methods operate on a fixed set of predefined tasks and require model retraining when new tasks arise. Furthermore, current approaches produce one-shot, static outputs, lacking the capacity for iterative refinement or deeper exploration of predictions. This position paper argues that multi-modal large language models (LLMs) are essential for enabling free-form and interactive prediction of biomolecular functions, and zero-shot generalization to new tasks without model retraining. These models can generate coherent and context-aware text outputs that reflect the complexity and nuance of diverse functional roles. Importantly, they can generalize to novel biomolecules whose functions are unknown or poorly characterized, and they enable generalization to new tasks through prompt-driven adaptation, eliminating the need for task-specific retraining. Additionally, multi-modal LLMs enable interactive, multi-turn dialogue, allowing users to iteratively refine queries, clarify contexts, and explore hypotheses in a dynamic and responsive manner. By leveraging these capabilities, multi-modal LLMs provide a scalable, adaptable, and generalizable framework for advancing biomolecular function prediction and accelerating biological discovery.

[1]University of California San Diego [2]University of California, San Francisco [3]University of California, Berkeley. Correspondence to: Pengtao Xie <p1xie@ucsd.edu>.

*Proceedings of the 43$^{rd}$ International Conference on Machine Learning*, Seoul, South Korea. PMLR 306, 2026. Copyright 2026 by the author(s).

## 1. Introduction

DNAs, RNAs, and proteins are fundamental components of living systems, orchestrating the complex processes that sustain life. DNAs, composed of nucleotide sequences, serve as repositories of hereditary information, encoding instructions necessary for cellular function and organismal development. These instructions are transcribed into RNAs, which play multifaceted roles ranging from transmitting genetic information as messenger RNAs (mRNAs) to performing regulatory and catalytic functions in the form of non-coding RNAs and ribozymes. Proteins, synthesized from mRNAs through the process of translation, are intricate macromolecules with specific amino acid sequences that dictate their three-dimensional structures and biological functions. Proteins are involved in virtually all cellular activities, including enzymatic catalysis, signal transduction, immune response, and structural support. Together, these biomolecules form the backbone of life, enabling diverse processes that underpin growth, reproduction, and adaptation (Marcotte et al., 1999; Bepler & Berger, 2021; Watson et al., 2023; Listov et al., 2024; Kortemme, 2024). Understanding the functions of these biomolecules is essential for advancing biological research and driving innovations in translational applications. Accurate predictions of their roles can aid in identifying disease mechanisms, discovering novel therapeutic targets, engineering synthetic pathways, and developing precise tools for genome editing.

However, biomolecular function prediction presents significant challenges. These molecules are characterized by extraordinary diversity and complexity, with their functions often contingent on specific sequence motifs, secondary and tertiary structures, post-translational modifications, and interactions within their cellular context (Lee et al., 2007; Radivojac et al., 2013; Peled et al., 2016; Rives et al., 2021; Bileschi et al., 2022; Peng et al., 2021; Zou et al., 2019; Zhao et al., 2020; Zhang et al., 2019; Noviello et al., 2020). For instance, a single protein may participate in diverse biological pathways, acting as an enzyme, a structural component, or a signaling molecule depending on its cellular environment. Similarly, RNAs can serve as molecular scaffolds, regulate gene expression, or act as sensors of cellular stress, adding to the challenge of functional characterization.

Traditional computational approaches for predicting biomolecular functions, such as sequence alignment tools (Thompson et al., 1997; Galtier et al., 1996; Kumar et al., 2004; Do et al., 2005; Li et al., 2009) and rule-based methods (Faeder et al., 2005; 2009; Smith et al., 2012; Chylek et al., 2014; Harris et al., 2016), rely on predefined features and are often limited in their ability to generalize across diverse datasets. While effective for specific tasks, these methods struggle to capture the complex patterns underlying biomolecular interactions and activities. Recent advancements in deep learning have revolutionized this field, enabling the development of models that can process large-scale datasets of sequences, structures, and annotations to identify intricate relationships that were previously inaccessible (Rives et al., 2021; Elnaggar et al., 2021; Meier et al., 2021; Brandes et al., 2022; Hsu et al., 2022; Xu et al., 2023; Hayes et al., 2025). For example, models such as CLEAN (Yu et al., 2023) leverage deep learning architectures to predict enzyme functions with unprecedented precision, surpassing traditional tools like BLASTp (Altschul et al., 1990). Similarly, graph neural networks and attention-based models have demonstrated potential in mapping gene regulatory networks and predicting protein-protein interactions (Ryu et al., 2019; Wan & Jones, 2020; Gligorijević et al., 2021; Unsal et al., 2022; Wang et al., 2022; Zhou et al., 2022; Yu et al., 2023; Kulmanov et al., 2024; Zhang et al., 2024; Peng et al., 2021; Zou et al., 2019).

Despite these advancements, current deep learning methodologies exhibit critical limitations. Most models reduce biomolecular functions to predefined categories or hierarchical labels (Radivojac et al., 2013; Wan & Jones, 2020; Gligorijević et al., 2021; Zhou et al., 2022; Yu et al., 2023; Kulmanov et al., 2024; Zhang et al., 2024). While this categorization is practical, it oversimplifies the multifaceted and overlapping roles of biomolecules — for example, a single protein may act both as an enzyme and a structural component. The functions of proteins, RNAs, and genes are often complex, context-dependent, and intertwined in ways that traditional categorical outputs are ill-equipped to represent. Additionally, the field remains fragmented due to the reliance on specialized models for individual tasks, such as predicting enzymatic activity, RNA secondary structure, or protein-protein interactions. This task-specific model development increases computational resource requirements, complicates workflows, and inhibits the development of integrative, cross-functional insights (Peled et al., 2016; Gligorijević et al., 2021; Zhou et al., 2022; Wang et al., 2022).

To address these challenges, **we argue that multi-modal large language models are essential for enabling free-form, multi-task, and interactive prediction of DNA, RNA, and protein functions.** These multi-modal models take non-textual biomolecular sequences — such as DNA, RNA, or proteins — as inputs, along with natural language prompts that define the prediction task, and produce free-form textual outputs. Leveraging the generative and reasoning capabilities of large language models (Huang et al., 2022; Lightman et al., 2023; Dubey et al., 2024; Wang et al., 2024; Shao et al., 2024; Singhal et al., 2025), they can generate detailed, free-form descriptions of biomolecular functions. Unlike traditional classification-based approaches that limit biomolecular function prediction to discrete categories, LLMs can generate rich, descriptive outputs that capture the full complexity of molecular roles. These outputs may specify enzymatic activities, regulatory functions, interaction partners, and involvement in signaling or metabolic pathways — providing a nuanced representation aligned with how domain experts annotate molecular roles.

Moreover, multi-modal LLMs introduce a paradigm shift by enabling a unified and scalable approach to diverse prediction tasks within a single model architecture. Using task-specific prompts, these models can flexibly perform a wide range of tasks — such as classification, generation, and regression — without requiring retraining or the development of separate models. This prompt-based flexibility eliminates the overhead of maintaining multiple task-specific models and supports streamlined deployment across varied biomedical prediction scenarios, thereby reducing resource demands and enhancing scalability. For instance, a single prompt can elicit predictions of protein-drug interactions, RNA tertiary structure, or the identification of novel gene regulatory elements, with the model generating comprehensive, context-aware responses. Unlike previous multi-task approaches that depend on predefined task sets and require architectural modifications and retraining to accommodate new tasks, the prompt-driven LLM framework maintains a fixed architecture and model weights. New tasks can be introduced simply by modifying the input prompt, enabling rapid, zero-shot generalization to evolving biological prediction needs.

Another significant capability of these models is their ability to engage in interactive dialogue with users, enabling iterative refinement of predictions and the exploration of complex, context-dependent questions. This interactivity is especially important given that biomolecular functions can vary across tissues, developmental stages, and environmental conditions. Unlike static models, which provide fixed outputs, these models produce free-form responses that support follow-up queries, clarification of context, and exploration of hypothetical scenarios. This fosters deeper engagement and a more nuanced understanding of biomolecular behavior, effectively positioning the model as a collaborative partner in hypothesis-driven discovery. For example, users can probe specific aspects of a protein's function or examine how it might behave under different biological conditions.

In summary, our central position is to advocate for a unified, generative, and interactive modeling paradigm for biomolecular function prediction. We move beyond rigid task-specific classifiers and static outputs, proposing a single model that flexibly generates rich descriptions grounded in diverse molecular inputs. It is important to note that our focus is on biomolecules whose functions are unknown or poorly characterized - cases where no annotations exist in the literature and cannot be retrieved by any search engine. Our goal is not to retrieve known information, but to predict novel functions directly from inputs such as amino acid sequences, nucleotide sequences, and 3D structures.

**Conflict of Interest Disclosure** There are no conflict of interest.

## 2. Multi-modal LLM for Protein Function Prediction

In this section, we illustrate how multi-modal LLMs can generate detailed functional predictions for proteins based on their amino acid sequences. We construct a Protein-to-Text LLM (Figure 5 and Equation 3) that comprises an atom-level protein foundation model for learning protein representations and an LLM that analyzes these representations to generate text predicting the protein's function.

### 2.1. Atom-Level Protein Foundation Model

Understanding protein function requires modeling the physicochemical interactions that govern molecular behavior. Although sequence-based approaches (Brandes et al., 2022; Xu et al., 2023) have advanced functional annotation, they often miss key determinants of protein folding, stability, and binding. Many biological processes—including enzyme catalysis, allosteric regulation, and drug binding—depend on precise spatial and energetic constraints that cannot be inferred from sequence alone. Even subtle perturbations, such as post-translational modifications or point mutations, can markedly alter function by reshaping local molecular environments. Addressing these challenges requires *atom-level representation learning*, which models atomic coordinates, chemical bonds, electrostatic interactions, and solvent effects, enabling predictive models to capture both local perturbations and global structural constraints for improved functional annotation, variant effect prediction, and protein–ligand interaction modeling.

In this section, we describe an atom-level protein foundation model (ALPFM) that represents amino acids at atomic resolution. The ALPFM combines a graph neural network (GNN) (Wu et al., 2020) with a transformer (Vaswani et al., 2017) to capture both local chemical interactions and long-range dependencies. The GNN encodes atom-level features by modeling molecular graphs with atoms as nodes and

bonds as edges, enabling the learning of detailed geometries such as hydrogen bonding, van der Waals forces, and electrostatic interactions. These representations are then processed by a transformer to derive amino acid-level embeddings that capture global context across the protein sequence. The model can be pretrained on 65 million protein sequences from UniRef (Suzek et al., 2007), ensuring robust and generalizable representations. Related atom-level protein foundation models have been explored in (Zheng et al., 2024; Flam-Shepherd et al., 2023; Chu et al., 2024a; Lin et al., 2023).

### 2.2. Protein-to-Text LLM

Next, we describe a Protein-to-Text LLM, termed Protein-Chat, which takes an amino acid sequence as input and generates free-form text predicting protein function. ProteinChat consists of three components: the atom-level protein foundation model (ALPFM) described in Section 2.1, an adaptor module, and a large language model (LLM). The ALPFM encodes atomic interactions, spatial conformations, and physicochemical properties of the input protein. The adaptor module (Liu et al., 2024b; Xiao et al., 2024) maps these high-dimensional protein representations into the token embedding space of a pretrained LLM using dimensionality reduction, projection, and feature alignment, while preserving key biochemical attributes. The LLM then generates natural language descriptions of molecular activity, biological roles, and potential interactions, producing interpretable and context-aware functional predictions by synthesizing learned biochemical priors with contextual reasoning. ProteinChat can be trained using paired protein–text data. UniProt (Consortium, 2019b) provides approximately 0.57 million such pairs, where functional descriptions are derived from experimentally validated findings reported in the literature. Alternative protein-to-text LLM architectures have been explored in (Wang et al., 2023d; Abdine et al., 2024; Liu et al., 2024a; Lv et al., 2024).

## 3. Multi-modal LLM for Gene Function Prediction

In this section, we illustrate how multi-modal LLMs can generate detailed functional predictions for a gene based on its DNA sequence. We construct a Gene-to-Text LLM (Figure 6 and Equation 4) comprising a global-local DNA language model that extracts representations from the input DNA sequence, an adaptor network that maps these representations to the latent space of a textual LLM, and a textual LLM that generates descriptive text about the gene's function.

## 3.1. Global-Local DNA Language Model

Accurate gene function prediction requires modeling both global and local nucleotide dependencies. Global dependencies capture long-range regulatory interactions involving elements such as enhancers, silencers, and promoters that influence gene expression across large genomic distances, while local dependencies—such as transcription factor binding sites, splice junctions, and short sequence motifs—govern nucleotide-level functional outcomes. Together, these interactions determine gene regulation and function, and modeling both scales enables a more complete understanding of gene activity within its genomic context.

In this section, we describe a global–local DNA language model that integrates HyenaDNA (Nguyen et al., 2024b) and DNABERT (Ji et al., 2021) to capture complementary representations for each nucleotide. Global representations are obtained using HyenaDNA, a genomic foundation model pretrained on the human reference genome with context lengths of up to one million tokens at single-nucleotide resolution, enabling efficient modeling of long-range dependencies. Local representations are derived from DNABERT, which uses a BERT (Devlin et al., 2019)–based architecture with k-mer tokenization to capture short-range nucleotide interactions. A fusion network integrates these global and local representations to jointly model nucleotide dependencies across scales. Pretraining can be performed using a global–local self-supervised learning framework that combines contrastive learning (Han et al., 2022) at the whole-gene level with masked nucleotide reconstruction on local subsequences, enabling the model to learn both long-range functional properties and fine-grained sequence features. Alternative DNA language model architectures have been explored in (Dalla-Torre et al., 2024; Nguyen et al., 2024a; Benegas et al., 2023; Sanabria et al., 2024).

## 3.2. Gene-to-Text LLM

Next, we illustrate how to construct a Gene-to-Text LLM that takes a gene's DNA sequence as input and generates text predicting its function. The model consists of three key components: a global-local DNA language model, an adaptor network, and a textual LLM. The global-local DNA language model first processes the input gene sequence by capturing both long-range dependencies and local sequence motifs. The extracted sequence representations are then passed through an adaptor network, which aligns them with the latent space of the textual LLM. This adaptor network, implemented as a lightweight transformer or a feedforward projection module, ensures compatibility between the DNA-derived embeddings and the token-based representations of the language model. Finally, the textual LLM, pretrained on biomedical corpora, generates descriptive text predicting the gene's function, incorporating prior biological knowledge

and contextual cues from similar genes. The National Center for Biotechnology Information (NCBI) database (Sayers et al., 2022) includes 0.23 million gene-function pairs spanning diverse organisms, providing a resource for training the Gene-to-Text LLM.

# 4. Multi-modal LLM for RNA Function Prediction

In this section, we illustrate how multi-modal LLMs can generate detailed functional predictions for RNAs. RNA function is determined not only by its nucleotide sequence, which encodes critical motifs such as binding sites for proteins, small molecules, and other RNAs, but also by its three-dimensional structure, which governs interactions essential for its activity. While sequence defines the linear arrangement of nucleotides, structural folding into secondary and tertiary conformations enables key interactions, including base pairing, stacking, and hydrogen bonding. These structural features are crucial for processes such as ribozyme catalysis, RNA-binding protein recognition, and regulatory stability. Integrating sequence and structural data allows models to capture both primary sequence determinants and spatial conformations, improving predictions of how mutations affect structural integrity and function or how structural motifs contribute to regulatory roles.

We construct an RNA-to-Text LLM (Figure 7 and Equation 5) that integrates multiple components: an RNA language model that extracts contextual representations of nucleotides from the input sequence, a 3D structure prediction model that infers the RNA's 3D structure from its sequence, a 3D structure encoder that extracts structural representations, and a textual LLM that synthesizes both sequence and structure representations to generate a textual prediction of the RNA's function.

## 4.1. RNA Language Model

In this section, we describe a bidirectional RNA language model based on Mamba (Gu & Dao, 2023) that efficiently encodes RNA sequences in both forward and reverse directions. Built on Selective State Spaces (SSMs) (Gu et al., 2021), Mamba effectively models long-range dependencies, making it well suited for capturing complex RNA sequence patterns. Bidirectional processing allows the model to integrate contextual information from both the 5' to 3' and 3' to 5' orientations, which is essential for modeling RNA structural and functional properties and for accurately predicting RNA functions. The model can be pretrained using a multi-task self-supervised learning (SSL) framework that combines complementary objectives, including masked nucleotide prediction, sequence ordering, and context prediction. Large-scale pretraining can leverage RNAcentral (Consortium, 2019a), which contains 42 million RNA sequences

from diverse species totaling 30 billion nucleotides. Alternative RNA language model architectures have been explored in (Chu et al., 2024b; Outeiral & Deane, 2024; Wang et al., 2023c; Penić et al., 2024; Zou et al., 2024).

## 4.2. RNA 3D Structure Prediction Model

Next, we illustrate how to construct a Mamba-based sequence-to-structure architecture for RNA 3D structure prediction, leveraging Mamba's advanced capability to model long-range dependencies within sequences. The RNA language model described in Section 4.1 forms the backbone of this architecture, encoding input sequences to capture both global context and local sequence motifs. These sequence encodings are further processed by a graph-structured Mamba decoder, which transforms the linear sequence data into a graph-based representation. This representation enables the model to account for complex spatial relationships and molecular interactions that define RNA's three-dimensional conformation. Graph-based decoding is particularly suited for RNA structures, as it captures critical elements such as base-pair interactions, stacking conformations, and tertiary motifs. The model can be trained on approximately 2,300 sequence-structure pairs sourced from RNA 3D Hub (Leontis & Zirbel, 2012), a comprehensive and curated repository of RNA 3D structures and annotations. Alternative architectures for RNA 3D structure prediction have been explored in (Wang et al., 2023b; Shen et al., 2024; Li et al., 2023; Pearce et al., 2022).

## 4.3. RNA-to-Text LLM

Next, we illustrate how to construct an RNA-to-Text LLM that takes an RNA sequence as input and generates text predicting its function. The model consists of four primary components: an RNA language model, a 3D structure prediction model, a 3D structure encoder, and a textual LLM. The RNA language model processes the input nucleotide sequence and generates contextual embeddings that capture sequence dependencies. Simultaneously, the 3D structure prediction model infers the RNA's tertiary conformation based on its nucleotide sequence. The resulting 3D structure is then processed by a structure encoder that extracts geometric, topological, and physicochemical features, encoding interactions such as base stacking, hydrogen bonding, and tertiary contacts. To integrate these representations, a cross-modal fusion component aligns sequence-based and structure-based embeddings. The fused representation is then fed into the textual LLM to generate human-readable descriptions of the RNA's function. RNAcentral (Consortium, 2019a) contains approximately 47,000 RNA sequences paired with detailed textual descriptions of their functions, which can be used to train the RNA-to-Text LLM.

## 5. Discussion

### 5.1. Multi-modal LLMs Enable the Prediction of Biomolecules' Multifaceted Functions Through Detailed Text Generation

Multi-modal LLMs can generate coherent and comprehensive text to predict the diverse functions of biomolecules, capturing their detailed roles, interactions, and biological context in a manner akin to human expert descriptions. We conducted preliminary studies, using the ProteinChat model to generate free-form text predictions for the functions of 200 randomly selected proteins from Swiss-Prot (UniProtKB, 2024). Experimental evaluations (Figure 1a,b), including human expert assessments and automated metrics, showed that ProteinChat outperforms GPT-4 by over tenfold. The predictions in Figure 1c illustrate that, unlike previous methods that predict protein functions as discrete categories, ProteinChat generates cohesive and thorough natural language narratives about the diverse functions of proteins. Previous methods often fall short in capturing the complexity and nuance of protein functions, as they reduce these functions to simplistic categories. ProteinChat, however, generates rich, detailed descriptions that mirror the comprehensive analyses provided by human experts. This capability allows for a more holistic understanding of proteins, encompassing their intricate roles, interactions, and biological significance. By utilizing large language models, ProteinChat describes the multifaceted nature of proteins in a way that is both accessible and scientifically rigorous. This method enhances our understanding of individual proteins and facilitates insights into the broader biological systems they operate within.

### 5.2. Multi-Modal LLMs Enable Diverse Prediction Tasks Within a Unified Framework

Multi-modal LLMs can perform a wide array of prediction tasks using task-specific user instructions or questions described in natural language (i.e., prompts), eliminating the necessity of training separate models for each task. We conducted preliminary studies using the ProteinChat model for five prediction tasks, including predicting catalytic functions, ligand binding functions, coenzyme-enzyme interactions, biological processes, and cellular component compartmentalization. We do not need to retrain ProteinChat for each task. Instead, using the same fixed ProteinChat model, we design task-specific prompts to guide it in executing different tasks. The five prompts we designed are: 'What type of enzyme is this?', 'What ligand can this protein bind to?', 'What coenzyme does this enzyme interact with?', 'What biological process is this protein involved in?', and 'In which cellular component is this protein localized?'. Figure 2 presents the results, showing that ProteinChat significantly outperforms task-specific classifiers, each trained to per-

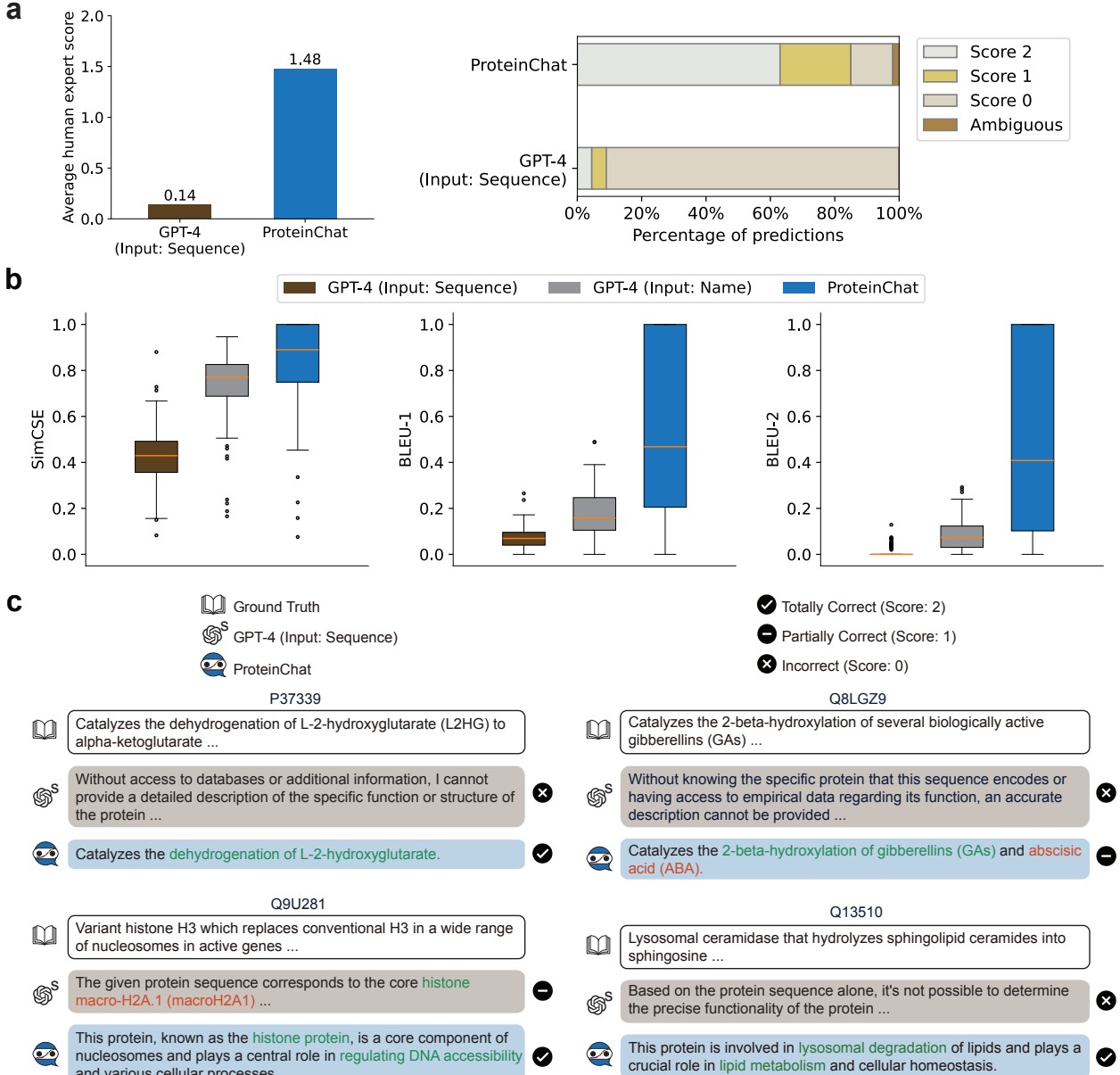

*Figure 1.* **ProteinChat accurately predicts protein functions expressed in textual descriptions and outperforms GPT-4**. **a**, ProteinChat significantly outperforms GPT-4 in human expert assessments, by more than ten-fold. Experts assessed predictions on a 0-2 scale: 2 for completely correct, 1 for partially correct, and 0 for incorrect. The average scores are on the left, with the distribution of scores on the right. Like ProteinChat, GPT-4 uses amino acid sequences of proteins as input. **b**, In automated evaluation metrics including SimCSE (Gao et al., 2021a), BLEU-1 (Papineni et al., 2002), and BLEU-2, ProteinChat demonstrates significantly superior performance compared to GPT-4 which uses amino acid sequences or protein names as inputs. **c**, Examples of predictions generated by ProteinChat and GPT-4 demonstrate that ProteinChat's predictions are more accurate and informative than those of GPT-4.

form a specific prediction task, and GPT-4 across various tasks. Developing a specialized model for each prediction task involves extensive training data collection, model training, and hyperparameter tuning, which is time-consuming, resource-intensive, and requires significant domain expertise to ensure accuracy and reliability. Additionally, specialized models cannot easily adapt to new or related tasks

without undergoing the entire development process again. In contrast, ProteinChat leverages a single model to perform a variety of protein function prediction tasks by simply modifying the prompts, thereby eliminating the need for developing separate models for each task. This enhances efficiency, flexibility, and scalability. Furthermore, we conducted additional comparisons of ProteinChat against other

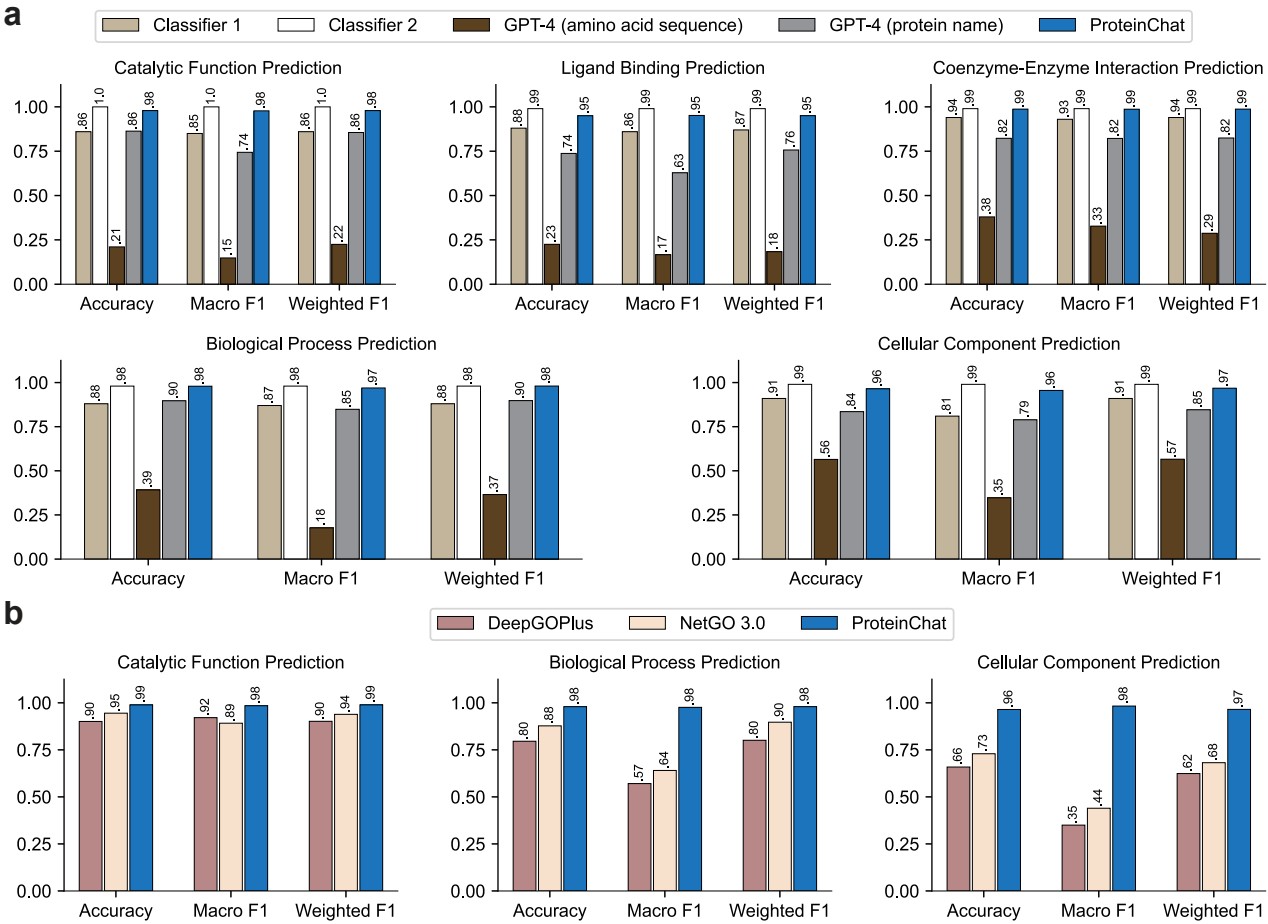

*Figure 2.* **ProteinChat significantly outperforms task-specific classifiers and GPT-4 across diverse prediction tasks. a**, In five prediction tasks curated from UniProt, including catalytic function prediction, ligand binding prediction, coenzyme-enzyme interaction prediction, biological process prediction, and cellular component prediction, ProteinChat achieves significantly better performance compared to specialized classifiers and GPT-4. Each classifier includes a protein encoder based on the pretrained xTrimoPGLM-1B (Chen et al., 2024) protein language model and a classification head based on a multi-layer perceptron. Given the amino acid sequence of a protein, the protein encoder extracts a representation vector which is subsequently fed into the classification head to predict the class label. For each classifier, we trained two variants: 1) keeping the pretrained protein encoder fixed and only training the classification head (referred to as Classifier 1), and 2) training both the protein encoder and the classification head (referred to as Classifier 2). **b**, In predicting protein functions represented using Gene Ontology (GO) (Consortium, 2004) categories, ProteinChat significantly outperforms two state-of-the-art GO classifiers - DeepGOPlus (Kulmanov & Hoehndorf, 2020) and NetGO 3.0 (Wang et al., 2023a).

state-of-the-art methods including ESM (Chen et al., 2024), Struct2GO (Jiao et al., 2023), SaProt (Su et al., 2024), and GearNet (Zhang et al., 2023b). The results are in Figure 4. ProteinChat outperforms these baselines, further highlighting the effectiveness of a single unified multi-modal LLM in handling a diverse range of protein function prediction tasks.

### 5.3. Multi-modal LLMs Enable Interactive and Iterative Predictions of Biomolecular Functions

Multi-modal LLMs facilitate interactive dialogues between users and the system. After obtaining the initial predictions, users can input more detailed and specific prompts to further refine and expand these predictions. We conducted prelimi-

nary studies using the ProteinChat model. Figure 3 in the appendix presents three example dialogues between Protein-Chat and human users, corresponding to proteins Q9U281, Q9XZG9, and Q9LU44 in UniProtKB. The dialogue on the left pertains to Q9U281, where the user inquires about the general function of this protein. ProteinChat identifies it as a histone protein involved in modulating DNA accessibility. Subsequently, the user inquires about the specific functions of this histone protein, and ProteinChat provides detailed predictions, highlighting the protein's roles in transcription regulation and post-translational modifications. The top right dialogue pertains to Q9XZG9, where ProteinChat initially predicts that the protein has antibacterial function. Based on the user's further prompt, ProteinChat then accurately predicts the protein can inhibit the growth of both

Gram-positive and Gram-negative bacteria. The bottom right example focuses on `Q9LU44`. When inquired about general functions, ProteinChat predicts that the protein is involved in pre-mRNA splicing. Upon further inquiry into specific molecular functions, such as metal binding, ProteinChat predicts that the protein binds zinc ions. This dynamic interaction between ProteinChat and users facilitates continuous, in-depth analysis of the same protein, in contrast to previous methods that offer only single-shot predictions. Users can delve deeper into the specifics of protein functions, exploring intricate details and nuances that single-shot predictions might miss. This ensures that the predictions are not only more accurate but also more comprehensive, uncovering complex protein behaviors and mechanisms.

## 6. Alternative Views

An alternative perspective to the position advocated in this paper is that task-specific deep learning models, rather than multi-modal LLMs, remain the more effective approach for biomolecular function prediction. This viewpoint argues that specialized models, trained on domain-specific datasets, achieve superior accuracy, interpretability, and reliability for individual prediction tasks. Proponents of this approach contend that multi-modal LLMs, while offering generalization across tasks, may struggle with domain-specific accuracy due to the inherent trade-offs in model optimization - particularly the risk of dilution in predictive power when attempting to generalize across multiple domains.

One primary concern is that multi-modal LLMs rely on natural language representations, which may not always encode the full biochemical and structural complexities necessary for precise biomolecular function prediction. In contrast, task-specific models directly leverage structured representations such as molecular graphs, energy landscapes, or sequence alignment scores, which have been optimized for capturing specific molecular properties. Additionally, while LLMs provide free-form textual outputs, these descriptions may introduce ambiguities or inconsistencies, making it harder to systematically validate predictions compared to the discrete outputs of specialized models.

To address these concerns, future developments in multi-modal LLMs should prioritize enhancing domain-specific accuracy through fine-tuned architectures and hybrid approaches that integrate structured representations alongside natural language generation. Additionally, rigorous benchmarking against specialized models will be essential to demonstrate that multi-modal LLMs can match or exceed their accuracy while offering added benefits such as flexibility and interactivity.

## 7. Call to Action

To translate the potential of multi-modal large language models into tangible advances in biomolecular function prediction, coordinated action is required across multiple stakeholders in the community. For model developers, we encourage the design of multi-modal LLMs that natively integrate biomolecular sequences, structures, and natural language prompts within a unified architecture. Priority should be given to prompt-driven task specification, enabling models to support diverse prediction objectives without task-specific retraining. Developers should also release standardized interfaces that facilitate interactive, multi-turn querying, allowing users to iteratively refine hypotheses and contextual assumptions during prediction. For dataset curators and benchmarking initiatives, we recommend the creation of evaluation protocols that go beyond single-shot classification accuracy. Benchmarks should assess a model's ability to generate coherent, biologically plausible free-form descriptions, adapt to previously unseen tasks via prompting, and support iterative refinement across multiple interaction rounds. Curated test cases involving poorly characterized or hypothetical biomolecules are particularly important to evaluate genuine predictive capability rather than information retrieval. For experimental biologists and domain experts, we advocate active participation in human-in-the-loop evaluation. Expert assessment of generated functional hypotheses, including plausibility, novelty, and experimental relevance, is essential for guiding model development and validating outputs. Close collaboration between model builders and experimentalists can help identify which forms of textual predictions are most actionable for downstream experimental design. For funding agencies and research organizers, targeted support is needed for interdisciplinary efforts that combine machine learning, molecular biology, and human–AI interaction. We encourage funding programs, shared tasks, and workshops that explicitly focus on interactive, multi-modal prediction paradigms. Collectively, these steps provide a concrete pathway for realizing the vision articulated in this position paper: moving from static, task-specific predictors toward interactive, generalizable, and hypothesis-generating systems for understanding the functions of DNAs, RNAs, and proteins.

## 8. Conclusions

This paper advocates for multi-modal large language models (LLMs) to advance biomolecular function prediction. Unlike traditional models constrained to predefined categories, multi-modal LLMs jointly process DNA, RNA, and protein sequences with natural language prompts to generate detailed, free-form functional descriptions. By integrating sequence, structure, and contextual information, these models enable more nuanced predictions, interactive refinement,

and unified support for diverse prediction tasks within a single framework, reducing reliance on task-specific models. These capabilities open multiple avenues for biological discovery. Multi-modal LLMs can predict functions for previously uncharacterized biomolecules from sequence and structural inputs alone, support hypothesis generation through interactive dialogue, and prioritize experimental targets by ranking predictions based on confidence or novelty. Future work should focus on improving interpretability, incorporating broader biological context, enhancing reliability, and validating predictions through experimental studies to enable robust biomedical applications.

## Acknowledgements

We acknowledge funding support from NSF IIS2405974, NSF IIS2339216, NIH R35GM157217, and NIH R21GM154171. We thank the reviewers for valuable feedback.

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

*Table 1.* Comparison of model performance based on SimCSE (Gao et al., 2021b) and BLEU (Papineni et al., 2002) scores for gene function prediction.

| Model | SimCSE | BLEU-1 | BLEU-2 | BLEU-3 | BLEU-4 |
|---|---|---|---|---|---|
| GPT-4 | 0.31 | 0.13 | 0.05 | 0.02 | 0.01 |
| Gene-to-Text LLM (ours) | **0.75** | **0.18** | **0.13** | **0.11** | **0.09** |

## A. Experimental Results for the Gene-to-Text LLM

We implemented and evaluated the Gene-to-Text LLM model described in Section 3 for gene function prediction. The model was trained on 0.23 million high-quality gene-function pairs from the National Center for Biotechnology Information (NCBI) database. Each input gene is represented by its nucleotide sequence, and the output function is expressed as free-form text extracted from scientific publications describing the gene's role. Preliminary results are presented in Table 1. Our method outperforms GPT-4 with a large margin.

## B. Experimental Details of ProteinChat

### B.1. Dataset preprocessing

We collected protein sequences and their functions from Swiss-Prot (UniProtKB, 2024), the reviewed subset of proteins in UniProtKB (Consortium, 2022). UniProtKB, a comprehensive database for protein sequences and functions, provides extensive information about the biological functions of proteins as derived from research literature. The "Function" section in UniProtKB gives a general overview of a protein's functions. Additionally, the "Keywords" section offers a controlled vocabulary with a hierarchical structure that describes various aspects of protein functions, including activities, locations, interactions, and more. The Swiss-Prot database within UniProtKB, which is manually curated by experts, serves as a high-quality reference for protein functions. The data used in this study is based on the UniProt 2023_02 version, released on May 2nd, 2023[1]. We downloaded the metadata in JSON format and extracted the protein functions by filtering entries where `commentType` is set to "Function". We excluded all functions with the `molecule` field, indicating that the function pertains to a subsequence after clipping rather than the entire protein sequence. This exclusion is necessary because the protein can serve as a precursor to various chains or peptides. UniProtKB specifies the role of each peptide separately under distinct `molecule`[2] entries. As a result, functions for 3,099 proteins were excluded, reducing the total to 522,966 proteins.

To optimize the training process, we limited the textual content of functions to a maximum of 400 characters. This constraint encourages concise summaries that effectively capture the function's essence. For longer texts, we used GPT-3.5 to condense the content into a single sentence summary with the following prompt: "Briefly summarize using one sentence started with *This protein*". We repeated the query until the response is under 400 characters, ensuring consistency in function text length.

We also generated (protein, prompt, answer) triples for training using UniProtKB keywords. The keywords for each protein can be structured using a directed acyclic graph, with a total of 10 first-level keywords. We selected 4 first-level keywords that describe protein functions, including molecular functions, binding properties, biological processes, and cellular localization. Additionally, we chose 31 second-level keywords that appear in over 10,000 proteins. These keywords cover 93.01% of all proteins in Swiss-Prot. After consolidating the keywords associated with each question, we developed six distinct prediction tasks. These tasks represent functions as discrete categories and are grouped into three main categories: Molecular Function, Biological Process, and Cellular Localization. These categories cover 67.1%, 35.5%, and 60.8% of all proteins, respectively. On average, each protein is involved in 2.7 tasks.

In our experiments, we randomly divided the proteins into training, validation, and test sets in proportions of 90%, 5%, and 5%, respectively. To maintain consistency, all designed triples related to a specific protein were grouped within the same set. For the test set, we randomly selected 200 proteins to evaluate free-form function prediction. Furthermore, we randomly selected 100 proteins for each specific prediction task. This resulted in a total of 800 proteins used for evaluation.

---

[1]https://www.uniprot.org/release-notes/2023-05-03-release

[2]https://www.uniprot.org/help/function

## B.2. Training details of ProteinChat

We used the Adam (Kingma & Ba, 2015) optimizer with $\beta_1 = 0.9$, $\beta_2 = 0.999$, and a weight decay of 0.05. We applied a cosine learning rate decay with a peak learning rate of $1e$-5 and a linear warm-up of 2000 steps. The minimum learning rate was $1e$-6. Due to the high memory consumption required for fine-tuning the encoder and LLM, we utilized a mini-batch size of one per GPU and limited the protein length to a maximum of 600 residues. Notably, 87.1% of the proteins had sequence lengths within this limit. For protein sequences longer than this limit, we truncated the excess length. We used 8 NVIDIA A100 GPUs, with 4 accumulation steps, resulting in an effective batch size of 32. To balance the sampling for different types of data, we used sampling rates of $1 : 2 : 10$ for keyword-based Q&A, rule-based Q&A, and manually annotated Q&A, respectively. We trained the model for 210K steps, covering a total of 1.6M Q&A pairs. In LoRA, we set the rank to 8, LoRA alpha to 16, and dropout rate to 0.05.

## B.3. Evaluation metrics

We employed SimCSE (Gao et al., 2021b) to assess the semantic similarity between the ground truth protein function and the predicted function. SimCSE leverages a contrastive learning framework (Hadsell et al., 2006) and utilizes the RoBERTa-base (Liu et al., 2019) model (denoted by $f_\theta$) to generate sentence embeddings. The semantic similarity is quantified by calculating the cosine similarity of these embeddings, with scores ranging from -1 to 1, where higher values signify greater semantic alignment. Specifically, let $s$ and $s'$ represent the ground truth protein function and the predicted function, respectively. The SimCSE score is computed as:

$$\cos_{\mathrm{sim}}(f_\theta(s), f_\theta(s')),$$

where $f_\theta(s)$ and $f_\theta(s')$ are the embeddings of $s$ and $s'$ extracted by the RoBERTa-base model $f_\theta$. $\cos_{\mathrm{sim}}(\cdot, \cdot)$ denotes the cosine similarity operation.

BLEU (Papineni et al., 2002) is computed using a set of modified n-gram precisions. Specifically,

$$\mathrm{BLEU} = \mathrm{BP} \cdot \exp\left(\sum_{n=1}^{N} w_n \log p_n\right) \tag{1}$$

where $p_n$ is the modified precision for n-gram, $w_n > 0$ and $\sum_{n=1}^{N} w_n = 1$. The brevity penalty (BP) is applied to penalize short generated text. Let $c$ be the length of the generated text and $r$ be the length of the ground truth. BP is computed as follows:

$$\mathrm{BP} = \left\{ \begin{array}{ll} 1 & \text{if } c > r \\ \exp(1 - \frac{r}{c}) & \text{if } c \leq r \end{array} \right\} \tag{2}$$

The weighted F1 score is computed by averaging the F1 scores of all categories, taking into account the number of true instances (support) for each category. The macro F1 score is calculated by averaging the F1 scores of all categories without considering their support. The macro F1 score is computed by taking the arithmetic mean (aka unweighted mean) of all the per-category F1 scores, and the weighted F1 score is calculated by taking the mean of all per-category F1 scores while considering each category's support.

In specific prediction tasks (i.e., classification tasks), both ProteinChat and GPT-4 occasionally produced responses containing multiple answers. For example, a response for biological process prediction might include both molecule transport and amino-acid biosynthesis. Such responses were deemed incorrect, even if they contained the correct answer. We only considered a response correct when it exclusively presented the single correct answer. Additionally, during the evaluation, all texts were standardized to lowercase to avoid the influence of letter casing.

## B.4. Experimental details for baseline methods

To solicit function predictions from GPT-4 using protein names, we used the following prompt: "You are a biologist specialized in protein functions. Given the name of a protein: [*protein name*], please describe the function of this protein in 1 or 2 sentences." When using the amino acid sequence of a protein to solicit function predictions from GPT-4, we used the following prompt: "Given the sequence of a protein: [*a string of amino acid letters such as MARYFRRRKFCRFTAEGVQEI-DYKDIATLKNYITESGKIVPSRITGTRAKYQRQLARAIKRARYLSLLPYTDRHQ*], please describe the function of this protein

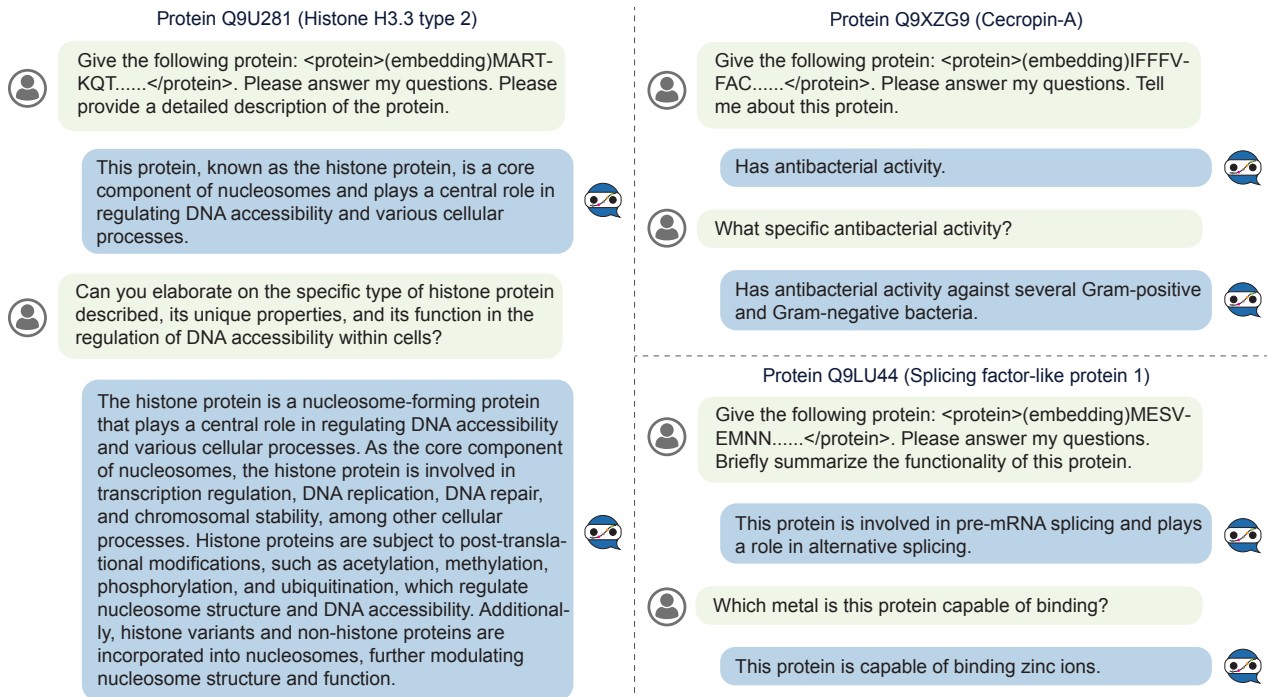

*Figure 3.* Interactive dialogues between ProteinChat and human users about proteins Q9U281, Q9XZG9, and Q9LU44.

in 1 or 2 sentences." We instructed GPT-4 to respond with 1 or 2 sentences to match the length of ground truth function descriptions in ProteinChat's training data, ensuring consistency.

Each specialized classifier consists of a protein encoder, specifically xTrimoPGLM-1B, and a classification head. Given the amino acid sequence of a protein, the protein encoder extracts representations for each amino acid. These representations are then averaged into a single vector, which is subsequently fed into the classification head to predict the class label. The classification head is a Multilayer Perceptron (MLP) with two layers. For all classification tasks, the first layer of the MLP contains 128 hidden units. The second layer's number of hidden units corresponds to the number of categories specific to the task. The weights of the MLP are initialized using the Kaiming initialization technique. We used the same learning rate and optimizer as in the ProteinChat training configurations. The batch size was set to 32, and a checkpoint was saved every 2500 iterations. The checkpoint with the best performance on 300 randomly selected validation examples was then chosen.

The two Gene Ontology (GO) classifiers - DeepGOPlus (Kulmanov & Hoehndorf, 2020) and NetGO 3.0 (Wang et al., 2023a) - utilize online web services to predict GO terms with rankings. A prediction is considered correct if the ground truth GO term holds the highest rank among all possible answers for the given question.

## C. Additional Figures and Equations

Figure 4 shows the experimental comparison between ProteinChat and state-of-the-art baselines in catalytic function prediction, ligand binding prediction, coenzyme-enzyme interaction prediction, biological process prediction, and cellular component prediction.

Figures 5, 6, and 7, as well as Equations 3, 4, and 5, illustrate the Protein-to-Text LLM, Gene-to-Text LLM, and RNA-to-Text LLM, respectively.

$$
\begin{aligned}
Z_s &= f_{alpfm}(S) \\
Z_a &= f_{adaptor}(Z_s) \\
p(T \mid Z_a) &= p_W\left(T_1 \mid Z_a\right) \prod_{i=2}^{L} p_W\left(T_i \mid Z_a, T_{<i}\right)
\end{aligned}
\tag{3}
$$

where $S$ denotes the amino acid sequence of an input protein. $f_{alpfm}$ denotes the atom-level protein foundation model. $Z_s$

denotes the representation of the protein. $f_{adaptor}$ denotes the adaptor. $Z_a$ denotes the adapted representation. For a target answer $T$ that has $L$ text tokens, the Protein-to-Text LLM computes the probability of generating $T$ in the third equation. We denote the $i$-th token as $T_i$ and all preceding tokens as $T_{<i}$. $W$ denotes the LLM's model weights.

$$
\begin{aligned}
Z_g &= f_{gldlm}(G) \\
Z_a &= f_{adaptor}(Z_g) \\
p(T \mid Z_a) &= p_W(T_1 \mid Z_a) \prod_{i=2}^{L} p_W(T_i \mid Z_a, T_{<i})
\end{aligned}
\tag{4}
$$

where $G$ denotes the nucleotide sequence of an input gene. $f_{gldlm}$ denotes the global-local DNA language model. $Z_g$ denotes the representation of the gene. $f_{adaptor}$ denotes the adaptor. $Z_a$ denotes the adapted representation. For a target answer $T$ that has $L$ text tokens, the Gene-to-Text LLM computes the probability of generating $T$ in the third equation. We denote the $i$-th token as $T_i$ and all preceding tokens as $T_{<i}$. $W$ denotes the LLM's model weights.

$$
\begin{aligned}
Z_r &= f_{rlm}(R) \\
S &= f_{spm}(Z_r) \\
Z_s &= f_{se}(S) \\
Z_{fusion} &= f_{fusion}(Z_r, Z_s) \\
p(T \mid Z_{fusion}) &= p_W(T_1 \mid Z_{fusion}) \prod_{i=2}^{L} p_W(T_i \mid Z_{fusion}, T_{<i})
\end{aligned}
\tag{5}
$$

where $R$ denotes the nucleotide sequence of an input RNA. $f_{rlm}$ denotes the RNA language model. $Z_r$ denotes the representation of the RNA. $f_{spm}$ denotes the structure prediction model. $S$ denotes the predicted structure. $f_{se}$ denotes the structure encoder. $Z_s$ denotes the representation of the 3D structure. $f_{fusion}$ denotes the cross-modal fusion component. $Z_{fusion}$ denotes the fused representation. For a target answer $T$ that has $L$ text tokens, the RNA-to-Text LLM computes the probability of generating $T$ in the third equation. We denote the $i$-th token as $T_i$ and all preceding tokens as $T_{<i}$. $W$ denotes the LLM's model weights.

## D. Applications

In this section, we present several potential applications of the multi-modal LLM described in previous sections.

### D.1. Predict Antibiotic and Immunomodulatory Functions of Proteins and Peptides

Numerous host-derived small proteins or peptides modulate the immune response to infectious disease and play a central role in dictating clinical outcome. These include proteins that exhibit direct antimicrobial activity (e.g., defensins, cathelicidins), others that recruit immune cells to the site of infection (chemokines such as CXCL8), and those that influence immune cell activation, performance, and cell death regulation (cytokines like IL-1, IL-6 and TNF). Exogenous microbial or plant-derived proteins and peptides, as well as synthetic versions, have likewise been developed for their antimicrobial or immunomodulatory action. The Protein-to-Text LLM can be applied to predict potential novel antibiotic or immunomodulatory functions of proteins and peptides. The translational potential of these predictions for human medicine can be evaluated using well-validated assays for: (1) antimicrobial susceptibility testing of direct bactericidal activity against leading human pathogens, including contemporary multidrug-resistant strains (e.g., MRSA, *Pseudomonas aeruginosa*, *Acinetobacter baumannii*); and (2) beneficial activities during bacterial interactions with immune components, such as serum, neutrophils, and macrophages, determining if the peptides/proteins improve immune cell bactericidal activity or halt pathological inflammatory responses. Readouts include macrophage and neutrophil oxidative burst, phagocytosis, apoptosis, pyroptosis, degranulation, chemotaxis/migration, cytokine release, and extracellular trap formation. This information can guide studies of therapeutic efficacy in murine models of pneumonia and sepsis using the corresponding pathogens. Such new drug discovery is desperately needed given the ever-expanding antibiotic resistance crisis and the impact of uncontrolled inflammation to increase morbidity and mortality in patients with pneumonia and/or sepsis.

### D.2. Predict Functions of Underannotated Genes

Metabolic network reconstructions are genome-scale knowledgebases of gene function assembled through manual curation of painstaking biochemical enzyme characterization experiments. These reconstructions have been extended to include Transcription, Translation, and Stress Response Machinery and now represent over 80% of the proteome by mass in the model organism *E. coli*. However, approximately one third of *E. coli* genes are still unannotated or only partially annotated.

The Gene-to-Text LLM can be used to predict potential functions for these underannotated genes. We can experimentally evaluate the accuracy of these predictions by performing gene knockouts and assessing their effects using assays relevant to the predicted gene functions, such as stress response or substrate catabolism. Furthermore, we can clone different LLM-suggested versions of an enzyme into *E. coli* and determine their relative activity levels through growth screens tied to the gene function in a similar fashion.

### D.3. Predict Functions of RNAs

RNA function prediction aims to uncover RNA's roles in cellular regulation, shedding light on disease mechanisms. Dysregulated RNA function is associated with conditions like Alzheimer's, autism spectrum disorders, and amyotrophic lateral sclerosis (ALS). The RNA-to-Text LLM can be applied to predict RNA functions, focusing on splicing, translation, stability, and localization, to inform targeted therapies. The LLM can identify functional motifs across transcriptomes, highlighting regions critical for RNA functions. Using an occlusion pipeline (Jin et al., 2023), tiled subsequences can be deleted to locate functional regions. High-throughput assays, such as spatial transcriptomics and CLIP-seq, can validate predictions, providing in vivo context. For example, the Xenium platform and immunofluorescence can analyze motifs in 5,000 transcripts. The pipeline in (Mah et al., 2024) can align predictions with experimental data for robust validation. To validate experimentally, RNA constructs with native sequences or motif-deleted variants can be tested using in situ hybridization and functional assays. Spatial transcriptomics can quantify changes caused by motif deletion, confirming regulatory roles. Therapeutically, validated motifs can guide RNA-based drug design for specific pathways with precision and minimal off-target effects. Scientifically, a map of RNA functional elements can enhance understanding of RNA-mediated regulation, advancing molecular biology and enabling discoveries in gene expression and cellular organization.

## E. Additional Discussions

### E.1. Scope and Novelty of the Proposed Framework

The novelty of this work does not lie in the mere use of multimodal models for biomolecular sequence function prediction, as prior studies (Garau-Luis et al., 2024; Tang et al., 2023; Zhang et al., 2023a; Mo et al., 2021) have already explored multimodal representations for specific biological tasks. Rather, the primary contribution of this paper is the position and unified framework it advocates for how biomolecular function prediction should be formulated and approached.

Existing multimodal approaches (Garau-Luis et al., 2024; Tang et al., 2023; Zhang et al., 2023a; Mo et al., 2021) typically focus on narrowly defined tasks, constrain outputs to predefined label spaces, and rely on task-specific architectures or retraining when new prediction objectives arise. As a result, these methods remain fragmented, with separate pipelines designed for different tasks such as enzyme classification, binding prediction, or functional annotation.

In contrast, this paper argues for a unified paradigm based on multi-modal large language models that fundamentally changes how biomolecular predictions are produced and used. Specifically, the proposed framework emphasizes three capabilities that are not jointly articulated in prior work: (i) zero-shot generalization to new tasks through prompt-based conditioning, eliminating the need for task-specific retraining; (ii) free-form textual prediction that captures the multifaceted, context-dependent nature of biomolecular functions beyond fixed categorical labels; and (iii) interactive, multi-turn prediction that enables iterative refinement, clarification, and hypothesis exploration rather than one-shot static outputs.

To our knowledge, no existing work in biomolecular sequence function prediction has explicitly framed this combination of unified architecture, free-form generation, and interactivity as a generalizable modeling paradigm. While prior multimodal models (Garau-Luis et al., 2024; Tang et al., 2023; Zhang et al., 2023a; Mo et al., 2021) demonstrate the feasibility of integrating multiple biological modalities, they are typically designed as task-specific solutions. The present work instead advances a conceptual shift toward treating biomolecular function prediction as a flexible, prompt-driven, and interactive reasoning problem, enabled by multi-modal LLMs. This distinction is important not only at a technical level but also at a methodological level, as it reframes how new biological questions can be addressed without repeatedly designing and training new models.

### E.2. Rationale for Multi-Turn Free-Text Interaction Beyond Decision Trees

The motivation for advocating multi-turn, dialogue-based interaction is rooted in the nature of biomolecular function prediction itself. Functional characterization often requires iterative refinement, contextualization, and hypothesis exploration

that cannot be exhaustively anticipated or encoded in a fixed decision tree. While decision trees can effectively handle predefined branches and outcomes, they inherently assume that all relevant questions, conditions, and response types are known in advance.

In practice, biological inquiry is open-ended. Domain experts frequently pose questions that evolve dynamically based on intermediate findings, including unexpected functional roles, condition-specific behaviors, mechanistic hypotheses, or interactions that fall outside established ontologies. Capturing such exploratory reasoning would require a decision tree with an impractically large—and continually expanding—set of branches. By contrast, free-text interaction allows users to formulate queries flexibly and enables the model to respond with nuanced descriptions that reflect the multifaceted and context-dependent nature of biomolecular functions, such as protein multifunctionality or environment-dependent regulation.

Multi-turn dialogue further enables chaining and refinement of queries in a way that static pipelines cannot support. A user may begin with a broad question (e.g., the general function of a protein) and progressively probe more specific aspects, such as ligand specificity, organismal context, or functional differences across variants. At each step, the model adapts its responses based on the evolving context of the conversation, rather than following a pre-specified path. Replicating this behavior with decision trees would require enumerating an infeasible number of conditional branches and hand-designed transitions.

Taken together, multi-turn free-text interaction represents a more general and extensible paradigm than fixed decision trees of pre-trained models. Rather than replacing structured approaches, it complements them by supporting open-ended exploration, adaptive questioning, and hypothesis generation in domains where the space of relevant biological questions cannot be fully specified in advance.

### E.3. Evaluation of Free-Form Text Generation in High-Stakes Biomedical Applications

A natural concern with moving from structured, standardized outputs to free-form natural language generation is the challenge of evaluation, particularly in high-stakes application domains such as biomedicine. Rigorous and reliable evaluation of free-form predictions is essential for the responsible deployment of multi-modal LLMs in these settings. To address this, we advocate for a multi-pronged evaluation strategy that combines expert assessment, automated metrics, and hybrid protocols.

- **Expert Review as the Gold Standard**. Domain expert evaluation remains central to assessing the correctness, completeness, and biological plausibility of free-form functional predictions. This mirrors existing biological annotation practices, such as those used in resources like UniProt, where functional descriptions are inherently narrative and curated by experts. Expert review can capture subtle biological nuances, contextual dependencies, and multifaceted functional roles that are difficult to encode in predefined label spaces. In our preliminary studies, expert assessments consistently indicated that free-form predictions provided richer and more accurate characterizations of biomolecular function than classification-based baselines.

- **Automated Metrics for Scalability and Comparability**. Complementary to expert review, automated evaluation metrics enable scalable and reproducible benchmarking. Semantic similarity measures based on embedding models (e.g., SimCSE) and text overlap metrics such as BLEU provide quantitative signals of consistency between generated outputs and validated references. While such metrics cannot fully capture biological correctness on their own, they are valuable for large-scale screening, model comparison, and continuous performance tracking during development.

- **Hybrid Evaluation Protocols**. We envision layered evaluation frameworks in which automated metrics are used as an initial screening mechanism to identify low-quality, inconsistent, or ambiguous predictions, followed by targeted expert review of high-value or uncertain cases. This hybrid approach balances scalability with rigor, enabling systematic evaluation while ensuring that biologically meaningful judgments remain grounded in expert knowledge.

- **Implications for High-Stakes Applications**. Although categorical outputs are easier to score, they often fail to represent the multifaceted and context-dependent nature of biomolecular functions. Free-form predictions, when paired with careful evaluation protocols, can offer more informative and biologically relevant insights. Moreover, the interactive, multi-turn dialogue capabilities of LLMs provide an additional layer of validation: experts can probe specific aspects of a prediction, request clarifications, and iteratively refine hypotheses, a capability that is not available in one-shot classification settings.

In summary, while evaluating free-form natural language outputs introduces new challenges, it also aligns computational predictions more closely with how functional knowledge is represented and assessed in biological research. By combining expert assessment, automated metrics, and hybrid evaluation strategies, free-form predictions generated by multi-modal LLMs can be systematically and reliably evaluated, supporting their responsible use in high-stakes biomedical applications.

### E.4. Mitigating Hallucinations in Free-Form Biomolecular Function Prediction

The free-form functions predicted by multi-modal LLMs may contain hallucinations — outputs that appear plausible but are not grounded in underlying biological evidence. To mitigate this issue, several strategies can be pursued. First, post-generation validation mechanisms can be incorporated. These may include confidence scoring, entailment-based consistency checks (Wu et al., 2023), or comparison against structured prediction outputs (e.g., GO (Consortium, 2004) terms) produced by parallel classifiers. Such strategies can flag or filter potentially unreliable responses, improving reliability without sacrificing the flexibility of free-form output. Second, interactive usage can serve as an additional safeguard. Because the model supports multi-turn dialogue, users can iteratively refine and probe predictions - for example, by asking for supporting evidence or clarification - creating an opportunity to detect and correct hallucinated content through user-driven inspection.

### E.5. Interpretability and Expert-Guided Use of Multi-Modal LLMs

Interpretability of predictions generated by multi-modal large language models (LLMs) is essential for ensuring that their outputs are scientifically meaningful, verifiable, and suitable for use in high-stakes biomedical applications. We recognize that LLM outputs can exhibit varying degrees of reliability, including uncertainty and occasional hallucinations. As a result, effective use of such models in scientific settings requires careful interpretation rather than blind acceptance of their predictions. In our view, domain experts should be positioned not as passive consumers of LLM outputs, but as active collaborators in validating, refining, and contextualizing model-generated hypotheses.

Interpretability can be enhanced through hybrid evaluation strategies that bridge free-form generation and structured validation. For example, generated text can be decomposed into structured assertions—such as predicted catalytic activities, cellular localizations, or interaction partners—which can then be cross-checked against curated databases or subjected to targeted experimental validation. In our preliminary evaluations, we combined expert assessment with automated comparisons to literature-derived annotations. These analyses showed that the generated outputs not only align with known biology, but often extend beyond existing categorical annotations in informative and biologically meaningful ways.

Transparency of predictions is a key component of informed interpretation. Our framework emphasizes presenting free-form outputs together with auxiliary signals, such as confidence estimates, attribution to specific molecular features or input regions, and ranked alternative hypotheses. These contextual cues enable experts to better judge the reliability of a prediction and to identify cases that warrant closer scrutiny or experimental follow-up.

We also emphasize the importance of expert training and interpretation guidelines. We envision providing domain experts with systematic guidance, analogous to annotation and curation protocols used in established resources such as UniProt. Such guidelines can highlight common failure modes of LLMs, including hallucinations and overgeneralization, and encourage best practices such as cross-checking predictions against prior biological knowledge, assessing experimental feasibility, and explicitly accounting for uncertainty when prioritizing hypotheses.

Interpretability is further enhanced through interactive validation via dialogue. The multi-turn interaction capability of multi-modal LLMs allows experts to iteratively probe and challenge initial predictions—for example, by requesting clarification, asking for supporting rationale, or exploring alternative hypotheses. This interactive process makes the model's reasoning more transparent and enables experts to "stress test" predictions in ways that are not possible with static, one-shot classifiers.

Importantly, we emphasize the complementary role of LLMs in the scientific discovery process. Predictions generated by LLMs are not intended to replace expert judgment or experimental validation. Rather, they serve as tools to accelerate hypothesis generation, highlight plausible functional candidates, and prioritize targets for downstream experimental investigation.

By coupling transparent model outputs with informed, structured, and interactive expert interpretation, uncertainty and hallucination risks can be managed in a principled manner. This human-in-the-loop perspective ensures that free-form predictions generated by multi-modal LLMs enhance—rather than undermine—scientific discovery in high-stakes biomedical domains.

## E.6. Computational Cost and Efficiency Trade-offs

The unified multi-modal LLM framework discussed in this paper does not aim to minimize computational cost in an absolute sense. Pre-training large multi-modal models, particularly those involving multi-stage pipelines such as the RNA-to-Text LLM, is inherently resource-intensive. The efficiency claim instead concerns reducing the long-term cost and complexity associated with developing, retraining, and maintaining many task-specific models.

In conventional biomolecular modeling pipelines, each new prediction task—such as RNA function annotation, structure-aware analysis, or interaction prediction—typically requires training a dedicated model, maintaining separate architectures, and repeating data curation and optimization procedures. By contrast, the unified framework consolidates these capabilities into a single model that supports diverse tasks through prompt-based adaptation. This consolidation avoids repeated model development cycles and reduces cumulative engineering, training, and maintenance overhead as tasks and benchmarks evolve.

In practice, the proposed multi-stage architectures are designed to reuse existing pretrained components. For example, the RNA-to-Text LLM leverages pretrained RNA language models and RNA structure prediction models as backbone components, with only lightweight adaptor modules trained to align these representations with a textual LLM. This design limits additional computational cost beyond the one-time pretraining of foundation models, while still enabling effective integration across modalities.

From a deployment perspective, serving a single unified model can also simplify system maintenance relative to managing a large collection of specialized models, each with its own inference pipeline and update schedule. Moreover, the prompt-driven design enables rapid zero-shot adaptation to new prediction targets or functional schemas without retraining, which is particularly advantageous in dynamic biological research settings.

Overall, while the upfront training cost of a large multi-modal LLM is substantial, the long-term efficiency gains arise from amortizing this cost across many tasks, reducing repeated model development, simplifying deployment, and enabling flexible adaptation as biological questions and datasets continue to evolve.

## E.7. Zero-Shot Generalization Beyond Classification Tasks

The unified framework advocated in this paper is not limited to classification-style predictions. Its central premise is that a single multi-modal large language model can support a broad spectrum of task types through prompt-based conditioning, without requiring task-specific retraining or architectural modification.

**Free-Form Prediction Beyond Classification.** Beyond question-based classification, the framework supports free-form generation of detailed functional descriptions. As demonstrated in Figure 1 and Section 5.1, the model generates rich textual predictions of protein functions that do not correspond to predefined categories. These outputs require synthesis and reasoning over molecular inputs and naturally extend beyond discrete label prediction.

**Extension to Quantitative Regression Tasks.** Prompt-based conditioning also extends naturally to quantitative regression. Because outputs are generated in free form, the same multi-modal LLM can be prompted to produce numerical values directly. For example, prompts can explicitly request predictions such as the catalytic efficiency of an enzyme or the binding affinity between a protein and a ligand (e.g., in kcal/mol). In these cases, the model generates numerical outputs rather than discrete category labels. This formulation effectively treats regression as a text generation problem over numerical tokens, allowing continuous and categorical predictions to be handled within a single, unified architecture. Importantly, no architectural modifications are required to support regression tasks: the same model can be prompted to produce free-text functional descriptions, structured assertions, or numerical estimates depending solely on the task specification provided in the prompt.

**Extension to Structure Generation.** The framework further accommodates generative tasks involving continuous geometric outputs. As described in Sections 4.2–4.3, RNA structure prediction is implemented by integrating a sequence-to-structure module within the multi-modal pipeline, enabling the generation of three-dimensional coordinates as part of the overall prediction process. This demonstrates that the framework can generalize beyond text-based outputs to tasks involving structured, continuous representations, while preserving the unified design.

Together, these examples illustrate that zero-shot generalization in the proposed framework extends beyond classification-

style tasks. By altering task prompts and leveraging modular multi-modal components, the same architecture can support descriptive generation, quantitative prediction, and structure generation within a unified model, avoiding the need to design and train separate architectures for different output modalities and providing a flexible foundation for diverse biomolecular prediction problems.

### E.8. Rationale for Using Mamba in RNA Sequence Modeling

Bidirectional encoding is not unique to Mamba and is also a property of transformer-based architectures. The motivation for adopting Mamba in the RNA sequence modeling component is therefore not bidirectionality per se, but computational efficiency and scalability for long nucleotide sequences.

RNA sequences, particularly when modeling long-range dependencies relevant to structure and function, can span tens to hundreds of thousands of nucleotides. Transformer-based models rely on pairwise self-attention, resulting in quadratic computational and memory complexity with respect to sequence length, which limits their practicality in such settings. In contrast, Mamba employs selective state space models with linear-time complexity, enabling efficient modeling of long RNA sequences while still capturing both local and global dependencies.

The discussion of bidirectional encoding in Section 4.1 is intended to emphasize that the RNA language model captures contextual information from both sequence directions, which is important for RNA function prediction, but this property should not be interpreted as the primary reason for selecting Mamba. Rather, the key advantage lies in its ability to scale efficiently to long sequences, making it well suited for RNA-centric tasks that are computationally challenging for attention-based architectures.

### E.9. Adapting to Evolving Biological Knowledge

The dynamic nature of biological databases poses an ongoing challenge for maintaining the relevance and accuracy of model predictions, as new discoveries continuously reshape the landscape of biological knowledge. To address this, techniques such as continual learning (De Lange et al., 2021), adapter-based fine-tuning (Hu et al., 2022), and retrieval-augmented generation (Lewis et al., 2020) can be leveraged to incorporate new information efficiently while preserving previously acquired knowledge. Our models can be updated incrementally as new annotated data becomes available, without needing to retrain from scratch. Moreover, the unified, prompt-based design of our models enables flexible adaptation to evolving tasks and biological contexts. As new prediction targets or functional ontologies emerge, the same model can be guided to address them using updated prompts and datasets - again avoiding the need to build and train task-specific models from the ground up.

### E.10. From Computational Feasibility to Experimental Validation

In this position paper, our primary aim is to articulate the conceptual and methodological case for multi-modal LLMs as a unified framework for biomolecular function prediction. While we present preliminary evidence demonstrating that such models can generate accurate and detailed predictions - validated by experts against annotations extracted from the scientific literature - we fully acknowledge that these computational results represent only the first step. Wet-lab experimental validation is a critical next stage. We have already identified high-confidence predictions from our model that are not present in existing annotations, and we are collaborating with experimental biologists to pursue their validation. Overall, generating and validating predictions at scale requires substantial time and resources, and we believe that establishing the methodological foundation and demonstrating computational feasibility is a necessary precursor to broad experimental validation efforts.

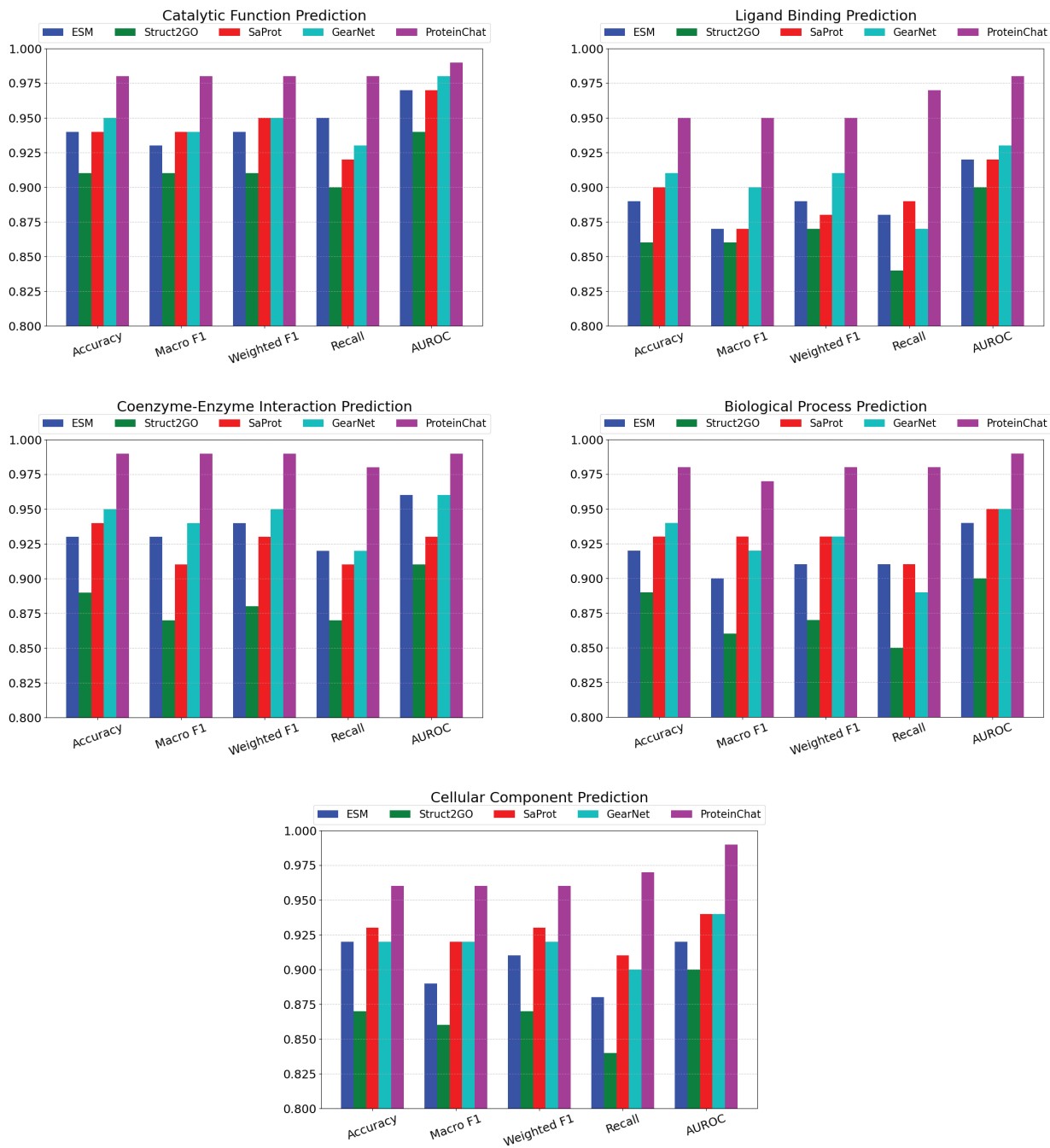

*Figure 4.* Compare ProteinChat with state-of-the-art baselines in catalytic function prediction, ligand binding prediction, coenzyme-enzyme interaction prediction, biological process prediction, and cellular component prediction.

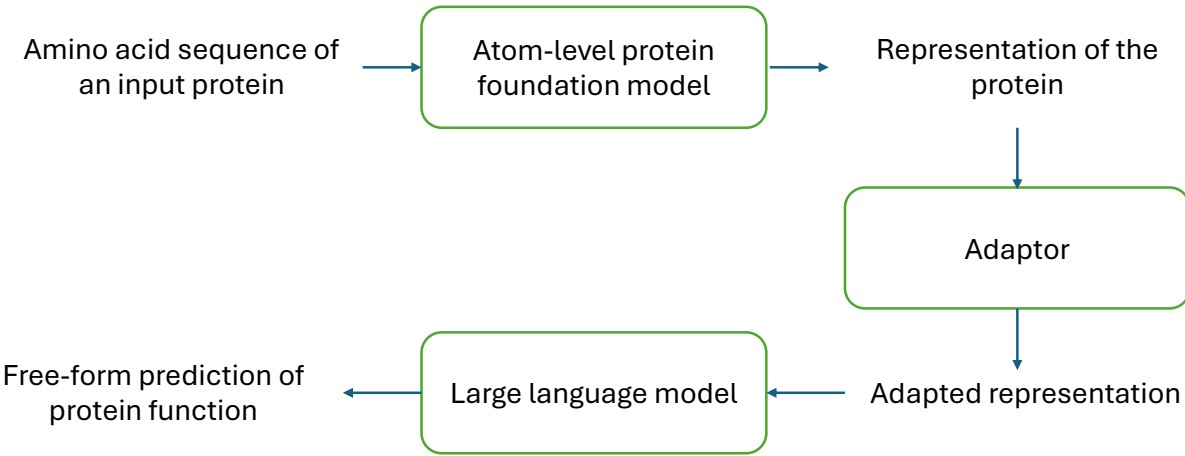

*Figure 5.* Illustration of the Protein-to-Text LLM

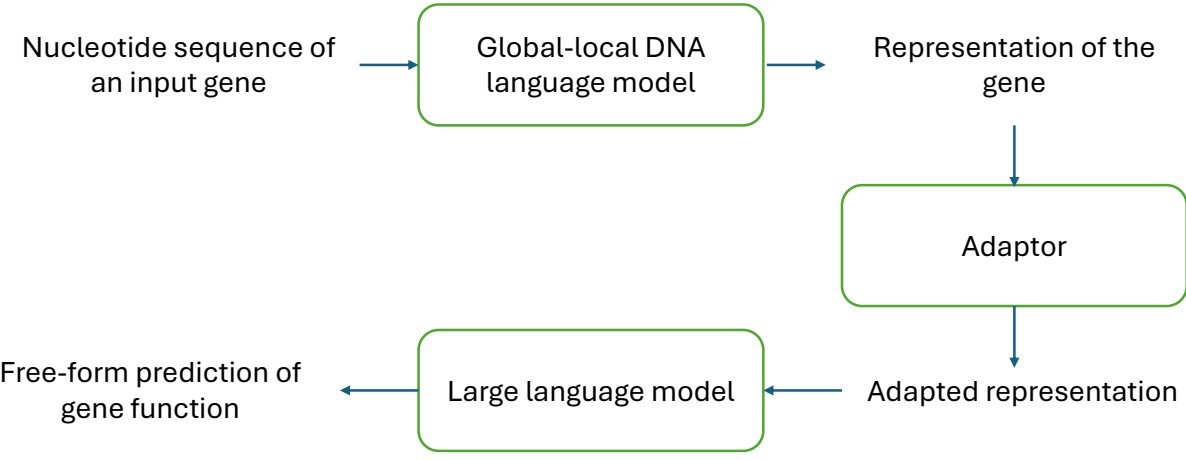

*Figure 6.* Illustration of the Gene-to-Text LLM

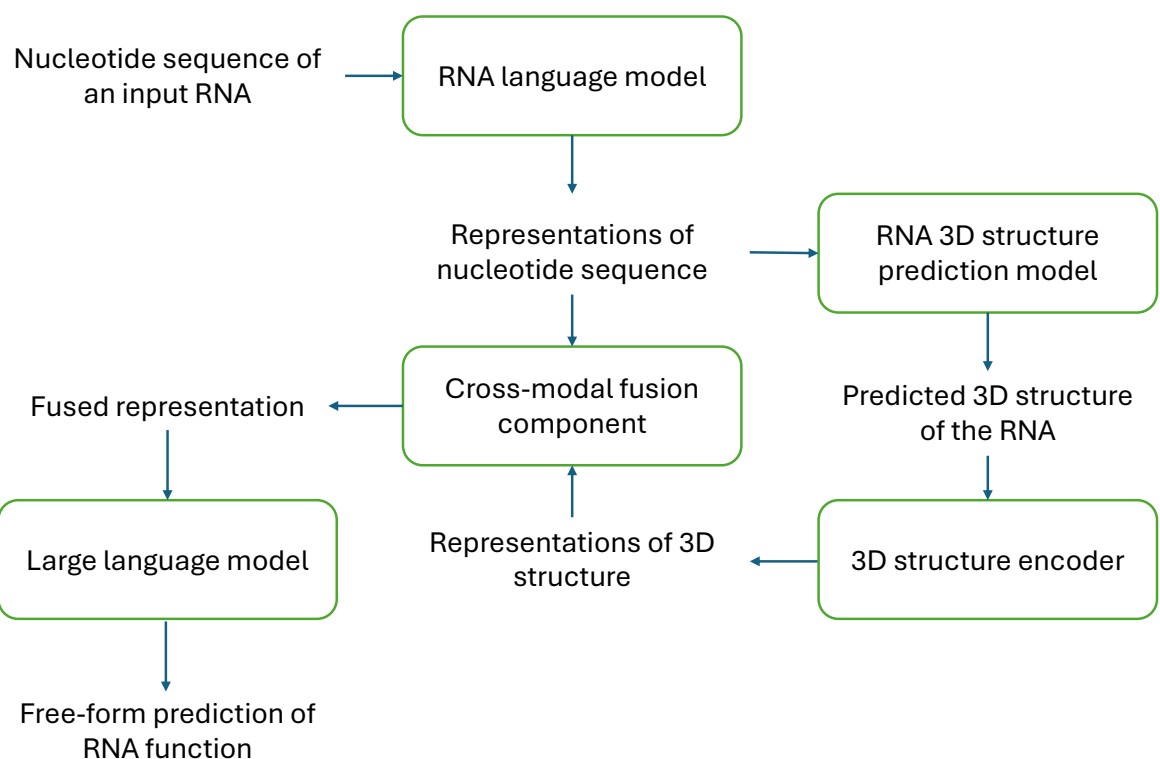

*Figure 7.* Illustration of the RNA-to-Text LLM

