# OpenReview forum: "Position: Deciphering the Functions of DNAs, RNAs, and Proteins Should Consider Multi-Modal Large Language Models"
_ICML.cc/2026/Position_Paper_Track — ICML 2026 Position Paper Track spotlight_

### Official Review · Reviewer_NhUC · 2026-03-07

**Significance:** 3
**Argument Clarity:** 4
**Rating:** 5
**Confidence:** 4

**Questions:**

1. I think it might be useful to have a subsection discussing the failure modes of LLM-driven biomolecular function prediction. It could be some format issues, significant hallucinations, etc, that are easily detectable and avoidable, but hard to notice (especially for the junior developers).

**Alternative Views Section:**

Yes

**Compliance With Llm Reviewing Policy A Conservative:**

Affirmed.

**Discussion Potential:**

3

**Final Justification:**

The rebuttal addressed my concerns. I would like to recommend the paper to appear at ICML.

**Paper Summary:**

This paper argues that biomolecular function prediction should move beyond traditional label-based classifiers toward multimodal large language models that generate textual functional descriptions. The paper discusses a unified framework that integrates molecular representations (such as sequences and structures) with LLMs to enable richer and more flexible function interpretation. Several examples from recent work are used to illustrate the potential of this paradigm.

**Position:**

Yes

**Position In Title:**

Yes

**Related Work:**

4

**Strengths And Weaknesses:**

Strengths:
1. I think it is a really impressive argument.
2. The paper is well-written and easy to follow.
3. The experiments are well-designed and nicely-presented.
4. The paper highlights the potential advantages of generative models in providing richer textual explanations and enabling interactive exploration, which may be valuable for hypothesis generation and biological interpretation.

Weaknesses:
I think there are minor weaknesses that would not affect the validity of the paper.
1. While the proposed paradigm is promising, the paper does not clearly discuss which types of biomolecular tasks would most benefit from generative LLM approaches compared with traditional prediction models.
2. There are also some important practical issues such as hallucination (which I think is really important and already affects the results a bit in some of my experiments), reliability of generated biological knowledge, and evaluation metrics for generated functional descriptions are only briefly discussed.

**Support:**

4

---

> ### Author Rebuttal · Authors · 2026-03-30
>
> We thank the reviewer for the positive and valuable feedback, which we will leverage to improve this work.
>
>
>
>
> ### **1. When Generative LLMs Provide the Greatest Benefit**
>
>
>
>
> We thank the reviewer for this valuable comment. In general, generative LLMs are particularly advantageous for tasks where **outputs are complex, multi-faceted, and not well captured by predefined labels**. Examples include:
> - **Functional annotation of poorly characterized biomolecules**, where functions are unknown or not covered by existing ontologies,
> - **Context-dependent function prediction**, where roles vary across conditions (e.g., tissue, environment, or interaction partners),
> - **Hypothesis generation and exploratory analysis**, where users seek open-ended insights rather than fixed predictions,
> - **Multi-aspect prediction**, where a molecule may simultaneously exhibit multiple roles (e.g., enzymatic activity, binding interactions, pathway involvement).
>
>
> In contrast, traditional models remain effective for well-defined tasks with structured outputs (e.g., classification or regression with fixed labels).
>
>
> We will revise the manuscript to explicitly discuss these distinctions and provide concrete examples to clarify when generative multi-modal LLMs offer the greatest benefit.
>
>
> $~$
>
>
>
> ### **2. Hallucination, Reliability, and Evaluation**
>
>
>
>
> We thank the reviewer for raising these important practical concerns.
>
>
>
>
> We agree that hallucination, reliability, and evaluation are critical challenges for applying generative LLMs to biomolecular function prediction, and we will expand the discussion in the revision.
>
>
>
>
> **2.1 Hallucination and reliability.**
> In the original manuscript, we discussed how to mitigate hallucinations in Appendix E.4 and improve reliability in Appendix E.5. We acknowledge that generative models may produce unsupported or inaccurate statements. To mitigate this, we can adopt strategies such as:
> - grounding predictions in **multi-modal inputs** (e.g., sequence and structure),
> - incorporating **external evidence** (e.g., databases or tool outputs),
> - and leveraging **uncertainty estimation** (e.g., confidence scores or self-consistency checks).
>
>
>
>
> These approaches can help improve trustworthiness and reduce unsupported predictions.
>
>
>
>
> **2.2 Evaluation of generated functions.**
> In the original manuscript, we discussed how to evaluate free-form text generation in Appendix E.3. We agree that evaluating free-form functional descriptions is non-trivial. In the revision, we will further clarify evaluation strategies, including:
> - **text similarity metrics** (e.g., embedding-based similarity),
> - **mapping generated text to structured labels** (e.g., GO terms) for quantitative evaluation,
> - and **expert assessment** for biological correctness and completeness.
>
>
>
>
> We will also discuss the limitations of each evaluation method.
>
>
>
>
> Overall, we will substantially expand the discussion of these issues to better reflect their importance and to clarify potential solutions. We appreciate the reviewer’s feedback and will incorporate it to strengthen the paper.
>
>
>
> $~$
>
>
> ### **3. Failure Modes of LLM-Based Biomolecular Prediction**
>
>
> We thank the reviewer for this valuable suggestion.
>
>
> We agree that explicitly discussing failure modes of LLM-driven biomolecular function prediction would strengthen the paper. In the revision, we will add a dedicated subsection covering common failure modes, including:
> - **Hallucinations** (unsupported or incorrect functional claims),
> - **Format inconsistencies** (e.g., incomplete or ambiguous outputs),
> - **Overgeneralization** from limited or noisy signals,
> - and **spurious or biologically implausible associations**.
>
>
> We will also highlight that some of these issues are **detectable and avoidable** (e.g., via consistency checks, validation against databases, or structural constraints), but may be overlooked—especially by less experienced users.
>
>
> In addition, we will discuss practical mitigation strategies, such as grounding predictions in multi-modal inputs, incorporating external tools or databases for verification, and using uncertainty estimation or self-consistency checks.
>
>
> We appreciate the reviewer’s suggestion and will incorporate this discussion to improve clarity and usability of the proposed framework.

---

> > ### Author Rebuttal · Reviewer_NhUC · 2026-04-01
> >
> > Thanks for the rebuttal; my concerns have been largely resolved. I think the paper deserves to appear in ICML.

---

### Official Review · Reviewer_gZCj · 2026-03-11

**Significance:** 2
**Argument Clarity:** 3
**Rating:** 3
**Confidence:** 3

**Questions:**

I am curious about your fusion algorithm: What algorithm is used to fuse LLM and protein embeddings, given you only have 47,000 pairs of examples as mentioned in section 4.3?

**Alternative Views Section:**

Yes

**Compliance With Llm Reviewing Policy A Conservative:**

Affirmed.

**Discussion Potential:**

2

**Final Justification:**

I have discussed with the authors.  My main concerns are regarding the technical details. Even though this is position track, if the authors have experiments, they still need be clear and reproducible.

**Paper Summary:**

This paper argues that DNA or protein fundation models should be designed as multimodal models, instead of specialized models for specific tasks.  Given the complexity and plurality of DNA / protein functions, training specialized models can be hard to generalize to other unseen scenarios and may neglect the multi-functionality of biological structures. The authors then propose a framework to combine the advantage of LLMs and specialized biology models; or more specifically, the protein sequence encoders. Some experiments are done to demonstrate that when equipped with LLMs, models are both generalizable and performing, often better than other specialized models and general-purpose models like ChatGPT.

**Position:**

Yes

**Position In Title:**

Yes

**Related Work:**

3

**Strengths And Weaknesses:**

The idea of this paper is crucial to the field of AI for biomedical research, as most of the research in this community focuses on single modality tasks due to the difficulty of data collection and training overhead. I agree that multimodality is an important step towards next-era biology AI models.

In addition to their arguments, the authors also propose and implement a framework to combine different encoders/decoders, including sequence encoder, structure predictor, structure encoder, and decoder-only LMs. Some experiments are done to demonstrate that their model is both generalizable and performing.

However, the argument of this paper is not new, nor controversial. For example, BioReason (Fallahpour et al, 2025) worked on a similar topic. Similar topics are also experimented on papers like EvoLLaMA and ProLLaMA, to name a few. The key challenge is to collect enough language-protein pairs, given that the alignment between 2 modalities usually requires at least million-size pairs.

Also, though the paper proposes and implemented a method, many technical details are missing. For example, what is the language model mentioned in section 4.3? How did you do alignment/fusion? Also, the description of some important components like structure encoder is limited to texts, without a clear definition or visualization. It is hard to judge or reproduce the results shown on this paper.

Missing reference:

Fallahpour, Adibvafa, Andrew Magnuson, Purav Gupta, et al. “BioReason: Incentivizing Multimodal Biological Reasoning within a DNA-LLM Model.” arXiv:2505.23579. 2025.


Update: I raised my rating from 2 to 3 during rebuttal.

**Support:**

2

---

> ### Author Rebuttal · Authors · 2026-03-30
>
> We thank the reviewer for the constructive and valuable feedback, which we will leverage to improve this work.
>
>
> ### **1. Clarification of Novelty, Scope, and Data Requirements**
>
>
> We thank the reviewer for the insightful comment and for highlighting related works such as BioReason (Fallahpour et al., 2025), EvoLLaMA, and ProLLaMA. We agree that recent efforts have begun exploring the integration of biomolecular representations with language models, and we appreciate the opportunity to clarify our contributions.
>
>
> **1.1 Distinction in scope and objective.**
> In the original manuscript, we clarified the novelty of our work in Appendix E.1. While prior works focus on aligning protein sequences with language models for improved performance on specific downstream tasks, they typically remain within **predefined task formulations** (e.g., classification or regression) and often require task-specific fine-tuning. In contrast, our work advocates a **unified, generative, and task-agnostic framework** that enables:
> - **Free-form functional prediction** beyond predefined ontologies,
> - **Prompt-driven task specification** without retraining,
> - **Interactive, multi-turn reasoning** over biomolecular functions.
>
>
> This shift—from task-specific prediction to **open-ended, language-based functional prediction**—is central to our position and, to our knowledge, has not been explicitly articulated in prior work.
>
>
> We emphasize that this paper aims to **present a conceptual and methodological direction**, rather than claim full novelty of the general idea. In the revision, we will better position our contributions relative to BioReason, EvoLLaMA, and ProLLaMA, clarifying both similarities and distinctions, and will add these works to the related work discussion.
>
>
>
> **1.2. On the challenge of obtaining language–protein paired data.**
> We agree that collecting large-scale paired data is an important challenge. Our work highlights that this challenge can be mitigated through:
> - Leveraging existing curated resources (e.g., UniProt, NCBI), which already provide substantial numbers of sequence–text pairs,
> - Utilizing self-supervised pretraining on large-scale unlabeled biomolecular data to learn strong representations prior to alignment,
> - Applying instruction tuning and prompt-based generalization, which reduce reliance on task-specific paired datasets.
>
>
> $~$
>
>
> ### **2. Clarification of Technical Details and Reproducibility**
>
>
> We thank the reviewer for this comment. In the original manuscript, we provided experimental details of ProteinChat in Appendix B. As this is a position paper, our focus is on articulating a conceptual framework and proof-of-concept implementation, rather than providing the full level of technical detail of a standard conference paper or a fully optimized system. That said, we agree that additional technical clarity would improve the paper, and we will address this in the revision.
>
> - In our experiments, we use RNA-FM [C1] as the RNA language model and Vicuna-13B as the textual LLM. We note that the framework is model-agnostic and can be readily extended to other RNA language models (e.g., UNI-RNA, AIDO.RNA) and textual LLMs (e.g., Qwen, GPT-OSS, LLaMA).
>
>
> - Our framework performs alignment via an adaptor module that projects biomolecular representations (sequence and/or structure embeddings) into the LLM token embedding space, followed by a cross-modal fusion mechanism (a lightweight transformer) that integrates sequence- and structure-level features. In the revision, we will provide architectural diagrams of the adaptor and fusion modules, precise formulations (e.g., projection and attention-based fusion), and implementation details (e.g., dimensions and training objectives).
>
>
> - In the revised version, we will explicitly describe the encoder architecture (e.g., graph-based or geometric over 3D coordinates), define its inputs (e.g., atomic coordinates, bonds) and outputs (structure embeddings), and provide a figure illustration for clarity.
>
>
> - To support full reproducibility, we will add more technical details into the current “Appendix B. Experimental Details of ProteinChat” section and release code.
>
>
> [C1] Shen et al. Accurate RNA 3D structure prediction using a language model-based deep learning approach. Nature Methods, 2024.
>
>
>
>
> $~$
>
>
>
> ### **3. Clarification of Fusion Mechanism**
>
> The fusion is implemented via a lightweight adaptor module that projects protein sequence and structure  embeddings into the token embedding space of the LLM, followed by cross-attention–based integration, where the LLM attends to these projected embeddings during text generation. This design enables effective alignment without requiring full end-to-end retraining. We will clarify these details in the revision. We also would like to clarify that the number of (protein, function text) pairs is 0.57 million, as described in Section 2.2. The 47k pairs in Section 4.3 are RNA-text pairs.

---

> > ### Author Rebuttal · Reviewer_gZCj · 2026-04-01
> >
> > Thanks for answering my questions and clarifying the number of protein-language pairs. As you would add more technical details, I agree to raise my rating.

---

### Official Review · Reviewer_MfP3 · 2026-03-12

**Significance:** 3
**Argument Clarity:** 3
**Rating:** 5
**Confidence:** 4

**Questions:**

No additional questions.

**Alternative Views Section:**

Yes

**Compliance With Llm Reviewing Policy A Conservative:**

Affirmed.

**Discussion Potential:**

3

**Final Justification:**

Rebuttal has addressed my concerns so I will keep my rate.

**Paper Summary:**

This paper argues that biomolecular function prediction should move to multi-modal LLM based approaches. This paper argues that such a movement could help generate context-aware text outputs, generalize to novel biomolecules better and enable human-model dialogue. Some initial experiments are presented to support the proposed position.

**Position:**

Yes

**Position In Title:**

Yes

**Related Work:**

3

**Strengths And Weaknesses:**

Strengths:
- This paper proposes a insightful position that could have broad impact to researchers in AI for biomedical science domain.
- Good supporting evidence, including initial experimental results, are provided to make the position sound.
- This paper is generally well written and organized.

Weaknesses:
- As LLM agents (LLM equipped with tools) is a dominating trend nowadays, I encourage authors to also discuss if multi-model LLM could benefit from domain-specifica tools for biomolecular function prediction.

**Support:**

4

---

> ### Author Rebuttal · Authors · 2026-03-30
>
> We thank the reviewer for the positive and valuable feedback, which we will leverage to improve this work.
>
>
> ### **1. Discussion on LLM Agents and Domain-Specific Tools**
>
> We thank the reviewer for this insightful suggestion.
>
> We agree that LLM agents (i.e., LLMs equipped with tools) are an important and rapidly growing direction, and that integrating domain-specific tools could further enhance multi-modal LLMs for biomolecular function prediction. While our current work focuses on a unified generative framework without external tools, we view tool integration as a natural and promising extension.
>
> Specifically, multi-modal LLMs can be augmented with domain-specific tools such as structure prediction, molecular docking, dynamics simulation, pathway analysis, and database retrieval. In such a framework, the LLM can act as a high-level controller that decomposes a query into sub-tasks, selects appropriate tools, and integrates their outputs. For example, given a protein sequence, the model may invoke a structure prediction module to obtain a 3D conformation, use docking tools to assess ligand binding, and query curated databases (e.g., UniProt or PDB) for homologous proteins and functional annotations. The outputs of these tools—such as binding affinities, structural motifs, and homologous evidence—can be fed back into the LLM, which synthesizes them into coherent functional predictions. This enables iterative refinement, where intermediate results guide subsequent tool selection and hypothesis generation.
>
> Importantly, tool integration can improve both accuracy and interpretability. External tools provide verifiable intermediate outputs (e.g., predicted structures or docking scores), helping ground predictions and mitigate hallucination. Incorporating uncertainty estimates (e.g., confidence scores from structure prediction) further enables calibrated outputs and identification of cases requiring additional analysis.
>
> From a systems perspective, this can be implemented through modular interfaces that connect the LLM with external APIs or executors, enabling dynamic tool invocation during inference. Future work may explore optimizing tool selection policies, efficient scheduling of expensive computations, and standardized integration of heterogeneous tools.
>
> We will include this discussion in the revision.

---

> > ### Author Rebuttal · Reviewer_MfP3 · 2026-04-03
> >
> > My questions have been addressed so I will keep my rate.

---

### Official Review · Reviewer_zp4r · 2026-03-12

**Significance:** 3
**Argument Clarity:** 3
**Rating:** 5
**Confidence:** 2

**Questions:**

Questions:
Please address the above weaknesses.

**Alternative Views Section:**

Yes

**Compliance With Llm Reviewing Policy A Conservative:**

Affirmed.

**Discussion Potential:**

3

**Final Justification:**

My concerns are addressed, thus I keep the posive score.

**Paper Summary:**

This paper proposes biomolecular function prediction should move from task-specific classifiers and fixed label spaces to multi-modal LLMs that produce free-form, interactive predictions across DNA/RNA/protein modalities.

**Position:**

Yes

**Position In Title:**

Yes

**Related Work:**

3

**Strengths And Weaknesses:**

Strengths:
1. The strucuture of the paper is coherent to build up a unified X-to-Text LLM template.
2. The authors include experiments to support their claim, enhancing the convince.

Weaknesses:
1. The comparison to GPT-4 is informative but inherently tricky since GPT-4 lacks domain-specific fine-tuning and some more advanced models are not involved into the comparison.
2. SimCSE and BLEU are weak proxies for biological correctness, so the evaluation should be more rigorous and comprehensive if the central claim is that free-text predictions are richer and more accurate than categorical ones.
3. The paper claims the framework can handle regression and other non-classification tasks via prompts, but presents no experiments.

**Support:**

2

---

> ### Author Rebuttal · Authors · 2026-03-30
>
> We thank the reviewer for the positive and valuable feedback, which we will leverage to improve this work. We will add the new experimental results and discussions into the revised paper.
>
> ### **1. Comparison with InstructProtein**
> We thank the reviewer for the valuable comment. Following the reviewer’s suggestion, we compared ProteinChat with InstructProtein [C1], a domain-specific, ChatGPT-style instruction-following model that accepts both natural language prompts and protein sequences as input and generates free-text predictions.
>
> As shown in the table below, ProteinChat substantially outperforms InstructProtein in human expert evaluation (average score: 1.48 vs. 0.52).
>
> | Model            | Average Human Expert Score |
> |------------------|----------------------------|
> | ProteinChat      | **1.48**                       |
> | InstructProtein  | 0.52                       |
> | GPT-4            | 0.14                       |
>
>
>
> The superior performance of ProteinChat over InstructProtein arises from fundamental architectural and pretraining differences. InstructProtein treats amino acids as ordinary text characters, mixing protein sequences with natural language during pretraining under a single next-token prediction objective. This approach fails to account for the distinct structural and statistical properties of protein sequences and limits its ability to model complex contextual and functional relationships. In contrast, ProteinChat employs a dedicated protein encoder pretrained on large-scale protein datasets, yielding richer, biologically grounded representations. By decoupling protein data from natural language and optimizing pretraining for biological sequences, ProteinChat attains a more accurate and nuanced understanding of protein functions.
>
> [C1] Wang et al. InstructProtein: Aligning human and protein language via knowledge instruction. ACL, 2024.
>
> $~$
>
>
> ### **2. Complementing Automatic Metrics with Human and LLM-Based Evaluation**
> We thank the reviewer for this important comment. In the original manuscript, as a complement to SimCSE and BLEU, we included human expert evaluation (Figure 1a), where domain experts assess the generated free-text predictions based on their alignment with ground-truth functional annotations. This provides a direct and biologically meaningful measure of prediction quality.
>
> To further strengthen the evaluation, we additionally incorporate an LLM-based assessment following the widely adopted LLM-as-a-judge paradigm. This approach enables scalable and systematic evaluation of free-form predictions and has been widely used in recent LLM research as a complement to human evaluation when expert annotation is costly and time-consuming.
>
> Specifically, for each generated prediction and its corresponding ground-truth annotation, we prompt Claude 3.5 Sonnet to assign a score using the same rubric as in the human evaluation. As shown in the table below, ProteinChat consistently outperforms the baselines under this evaluation protocol.
>
>
> | Model            | Average Claude Evaluation Score |
> |------------------|----------------------------|
> | ProteinChat      | **1.61**                       |
> | InstructProtein  | 0.73                       |
> | GPT-4            | 0.29                       |
>
> $~$
>
>
> ### **3. Evaluation on Regression Tasks**
>
> We thank the reviewer for this valuable comment. Following the reviewer’s suggestion, we evaluated ProteinChat on a regression task: predicting the minimum inhibitory concentration (MIC) of antimicrobial peptides.
>
> Given the amino acid sequence of an antimicrobial peptide, the goal is to predict its MIC, defined as the lowest concentration required to inhibit visible microbial growth. We focus on Escherichia coli (E. coli), a Gram-negative bacterium. We use the dataset from [C2], which contains 3,695 training examples and 924 test examples. Mean squared error (MSE) is used as the evaluation metric.
>
> ProteinChat handles this regression task via prompting to generate numerical predictions in free-text form. As shown in the table below, ProteinChat achieves lower prediction error than both GPT-4 and InstructProtein, demonstrating its ability to generalize beyond classification to continuous prediction tasks.
>
> | Model            | MSE |
> |------------------|----------------------------|
> | ProteinChat      | **0.49**                       |
> | InstructProtein  | 1.17                       |
> | GPT-4            | 1.62                       |
>
> [C2] Ledesma-Fernandez et al. Engineered repeat proteins as scaffolds to assemble multi-enzyme systems for efficient cell-free biosynthesis. Nature Communications, 2023.

---

> > ### Author Rebuttal · Reviewer_zp4r · 2026-04-03
> >
> > Thanks for the rebuttal. My concerns are addressed. I will keep the posive score.

---

### Decision · Program_Chairs · 2026-04-30

**Decision:**

Accept (spotlight)

**Comment:**

The dialogue during the rebuttal phase significantly clarified the paper's contribution and strengthened its evidence base:
•	Dedicated Encoders vs. Sequence-as-Text: The authors clarified a critical architectural insight—ProteinChat outperforms models like InstructProtein because it uses a dedicated protein encoder. Treating amino acids as ordinary text characters (the "sequence-as-text" approach) fails to capture the distinct structural and statistical properties of biological data.
•	Multi-Tiered Evaluation: To address concerns about weak metrics, the authors introduced a more robust evaluation protocol combining human expert scores and LLM-as-a-judge (using Claude 3.5 Sonnet). This demonstrated that ProteinChat significantly outperforms both general-purpose models (GPT-4) and domain-specific instruction models in biological grounding.
•	The "LLM Agent" Extension: An insightful discussion emerged regarding the trend of LLM agents. The authors agreed that while their current focus is a unified generative framework, the next step is using MLLMs as high-level controllers. These agents would invoke domain-specific tools (e.g., molecular docking, structure prediction) to ground their textual outputs in verifiable intermediate data, thereby mitigating hallucinations.
•	Generalization to Regression: The authors provided new experimental data on predicting minimum inhibitory concentrations (MIC), proving that their generative framework can effectively handle continuous numerical tasks via prompting.